# Constraining Electroweakinos in the
# Minimal Dirac Gaugino Model

Mark D. Goodsell[1★], Sabine Kraml[2†], Humberto Reyes-González[2‡],
Sophie L. Williamson[1,3§]

**1** Laboratoire de Physique Théorique et Hautes Energies (LPTHE), UMR 7589, Sorbonne
Université et CNRS, 4 place Jussieu, 75252 Paris Cedex 05, France.
**2** Laboratoire de Physique Subatomique et de Cosmologie (LPSC), Université
Grenoble-Alpes, CNRS/IN2P3, 53 Avenue des Martyrs, 38026 Grenoble, France
**3** Institute for Theoretical Physics, Karlsruhe Institute of Technology,
76128 Karlsruhe, Germany

★ goodsell@lpthe.jussieu.fr, † sabine.kraml@lpsc.in2p3.fr,
‡ humberto.reyes-gonzalez@lpsc.in2p3.fr, § sophie.williamson@kit.edu

## Abstract

Supersymmetric models with Dirac instead of Majorana gaugino masses have distinct phenomenological consequences. In this paper, we investigate the electroweakino sector of the Minimal Dirac Gaugino Supersymmetric Standard Model (MDGSSM) with regards to dark matter (DM) and collider constraints. We delineate the parameter space where the lightest neutralino of the MDGSSM is a viable DM candidate, that makes for at least part of the observed relic abundance while evading constraints from DM direct detection, LEP and low-energy data, and LHC Higgs measurements. The collider phenomenology of the thus emerging scenarios is characterised by the richer electroweakino spectrum as compared to the Minimal Supersymmetric Standard Model (MSSM) – 6 neutralinos and 3 charginos instead of 4 and 2 in the MSSM, naturally small mass splittings, and the frequent presence of long-lived particles, both charginos and/or neutralinos. Reinterpreting ATLAS and CMS analyses with the help of SModelS and MadAnalysis 5, we discuss the sensitivity of existing LHC searches for new physics to these scenarios and show which cases can be constrained and which escape detection. Finally, we propose a set of benchmark points which can be useful for further studies, designing dedicated experimental analyses and/or investigating the potential of future experiments.

# 1  Introduction

The lightest neutralino [1–3] in supersymmetric models with conserved R-parity has been the prototype for particle dark matter (DM) for decades, motivating a multitude of phenomenological studies regarding both astrophysical properties and collider signatures. The ever tightening experimental constraints, in particular from the null results in direct DM detection experiments, are however severely challenging many of the most popular realisations. This is in particular true for the so-called well-tempered neutralino [4] of the Minimal Supersymmetric Standard Model (MSSM), which has been pushed into blind spots [5] of direct DM

detection. One sub-TeV scenario that survives in the MSSM is bino-wino DM [6–9], whose discovery is, however, very difficult experimentally [10–12].

It is thus interesting to investigate neutralino DM beyond the MSSM. While a large literature exists on this topic, most of it concentrates on models where the neutralinos – or gauginos in general – have Majorana soft masses. Models with Dirac gauginos (DG) have received much less attention, despite excellent theoretical and phenomenological motivations [13–59]. The phenomenology of neutralinos and charginos ("electroweakinos" or "EW-inos") in DG models is indeed quite different from that of the MSSM. The aim of this work is therefore to provide up-to-date constraints on this sector for a specific realisation of DGs, within the context of the Minimal Dirac Gaugino Supersymmetric Standard Model (MDGSSM)

The colourful states in DG models can be easily looked for at the LHC, even if they are "supersafe" compared to the MSSM – see e.g. [47,58,60–71]. The properties of the Higgs sector have been well studied, and also point to the colourful states being heavy [38, 56, 59, 72–74]. However, currently there is no reason that the electroweak fermions must be heavy, and so far the only real constraints on them have been through DM studies. Therefore we shall begin by revisiting neutralino DM, previously examined in detail in [75] (see also [76,77]), which we update in this work. We will focus on the EW-ino sector, considering the lightest neutralino $\tilde{\chi}_1^0$ as the Lightest Supersymmetric Particle (LSP), and look for scenarios where the $\tilde{\chi}_1^0$ is a good DM candidate in agreement with relic density and direct detection constraints. In this, we assume that all other new particles apart from the EW-inos are heavy and play no role in the phenomenological considerations.

While the measurement of the DM abundance and limits on its interactions with nuclei have been improved since previous analyses of the model, our major new contribution shall be the examination of up-to-date LHC constraints, in view of DM-collider complementarity. For example, certain collider searches are optimal for scenarios that can only over-populate the relic density of dark matter in the universe, so by considering both together we obtain a more complete picture.

Owing to the additional singlet, triplet and octet chiral superfields necessary for introducing DG masses, the EW-ino sector of the MDGSSM comprises six neutralinos and three charginos, as compared to four and two, respectively, in the MSSM. More concretely, one obtains pairs of bino-like, wino-like and higgsino-like neutralinos, with small mass splittings *within* the bino (wino) pairs induced by the couplings $\lambda_S$ ($\lambda_T$) between the singlet (triplet) fermions with the Higgs and higgsino fields. As we recently pointed out in [69], this can potentially lead to a long-lived $\tilde{\chi}_2^0$ due to a small splitting between the bino-like states. Moreover, as we will see, one may also have long-lived $\tilde{\chi}_1^\pm$. As a further important aspect of this work, we will therefore discuss the potential of probing DG DM scenarios with Long-Lived Particle (LLP) searches at the LHC.

LHC signatures of long-lived Dirac charginos were also discussed in [78], albeit in a gauge-mediated R-symmetric model. The phenomenology of Dirac neutralinos and charginos at $e^+e^-$ colliders was discussed in [79].

The paper is organised as follows. In section 2 we discuss the EW-ino sector of DG models in general and within the MDGSSM, the focus of this work, in particular. This is supplemented by a comparative review of the Minimal R-Symmetric Standard Model (MRSSM) in appendix A.1. In section 3 we explain our numerical analysis: concretely, the setup of the parameter scan, the tools used and constraints imposed, and how chargino and neutralino decays are computed for very small mass differences. In particular, when the phase-space for decays is small enough, hadronic decays are best described by (multi) pion states (rather than quarks),

and we describe the implementation of the numerical code to deal with this. Furthermore, loop-induced decays of EW-inos into lighter ones with the emission of a photon can be important, and we describe updates to public codes to handle them correctly.

The results of our study are presented in section 4. We first delineate the viable parameter space where the lightest neutralino of the MDGSSM is at least part of the DM of the universe, and then discuss consequences for collider phenomenology. Re-interpreting ATLAS and CMS searches for new physics, we characterise the scenarios that are excluded and those that escape detection at the LHC. In addition, we give a comparison of the applicability of a simplified models approach to the limits obtained with a full recasting. We also briefly comment on the prospects of the MATHUSLA experiment. In section 5 we then propose a set of benchmark points for further studies. A summary and conclusions are given in section 6.

The appendices contain additional details on the implementation of the parameter scan of the EW-ino sector (appendix A.2), and on the identification of parameter space wherein lie experimentally acceptable values of the Higgs mass (appendix A.3). Finally, in appendix A.4, we provide some details on the reinterpretation of a 139 fb$^{-1}$ EW-ino search from ATLAS, which we developed for this study.

## 2 Electroweakino sectors of Dirac gaugino models

### 2.1 Classes of models

Models with Dirac gaugino masses differ in the choice of fields that are added to extend those of the MSSM, and also in the treatment of the R-symmetry. Both of these have significant consequences for the scalar ("Higgs") and EW-ino sectors. In this work, we shall focus on constraints on the EW-ino sector in the MDGSSM. Therefore, to understand the potential generality of our results, we shall here summarise the different choices that can be made in other models, before giving the details for ours.

To introduce Dirac masses for the gauginos, we need to add a Weyl fermion in the adjoint representation of each gauge group; these are embedded in chiral superfields **S**, **T**, **O** which are respectively a singlet, triplet and octet, and carry zero R-charge. Some model variants neglect a field for one or more gauge groups, see e.g. [28, 80]; limits for those cases will therefore be very different.

The Dirac mass terms are written by the *supersoft* [16] operators

$$\mathcal{L}_{\text{supersoft}} = \int d^2\theta \Big[ \sqrt{2}\, m_{DY} \theta^\alpha \mathbf{W}_{1\alpha} \mathbf{S} + 2\sqrt{2}\, m_{D2} \theta^\alpha \text{tr}\, (\mathbf{W}_{2\alpha} \mathbf{T})$$
$$+ 2\sqrt{2}\, m_{D3} \theta^\alpha \text{tr}\, (\mathbf{W}_{3\alpha} \mathbf{O}) \Big] + \text{h.c.}, \qquad (1)$$

where $\mathbf{W}_{i\alpha}$ are the supersymmetric gauge field strengths. It is possible to add Dirac gaugino masses through other operators, but this leads to a hard breaking of supersymmetry unless the singlet field is omitted – see e.g. [55]. On the other hand, whether we add supersoft operators or not, the difference appears in the scalar sector (the above operators lead to scalar trilinear terms proportional to the Dirac mass), so would not make a large difference to our results.

There are then two classes of Dirac gaugino models: ones for which the R-symmetry is conserved, and those for which it is violated. If it is conserved, with the canonical example being the MRSSM, then since the gauginos all carry R-charge, the EW-inos must be exactly

Chiral and gauge multiplet fields of the MSSM

| Superfield | Scalars | Fermions | Vectors | $(SU(3), SU(2), U(1)_Y)$ | $R$ |
|---|---|---|---|---|---|
| $\mathbf{H_u}$ | $(H_u^+, H_u^0)$ | $(\tilde{H}_u^+, \tilde{H}_u^0)$ | | $(\mathbf{1}, \mathbf{2}, 1/2)$ | $R_H$ |
| $\mathbf{H_d}$ | $(H_d^0, H_d^-)$ | $(\tilde{H}_d^0, \tilde{H}_d^-)$ | | $(\mathbf{1}, \mathbf{2}, \text{-}1/2)$ | $2 - R_H$ |
| $\mathbf{W_{3,\alpha}}$ | | $\lambda_3$ | $G_\mu$ | $(\mathbf{8}, \mathbf{1}, 0)$ | 1 |
| $\mathbf{W_{2,\alpha}}$ | | $\tilde{W}^0, \tilde{W}^\pm$ | $W_\mu^\pm, W_\mu^0$ | $(\mathbf{1}, \mathbf{3}, 0)$ | 1 |
| $\mathbf{W_{Y,\alpha}}$ | | $\tilde{B}$ | $B_\mu$ | $(\mathbf{1}, \mathbf{1}, 0\ )$ | 1 |

Additional chiral and gauge multiplet fields in the case of Dirac gauginos

| Superfield | Scalars, $R = 0$ | Fermions, $R = -1$ | $(SU(3), SU(2), U(1)_Y)$ |
|---|---|---|---|
| $\mathbf{O}$ | $O^a = \frac{1}{\sqrt{2}}(O_1^a + iO_2^a)$ | $\chi_O^a$ | $(\mathbf{8},\mathbf{1},0)$ |
| $\mathbf{T}$ | $T^0 = \frac{1}{\sqrt{2}}(T_P^0 + iT_M^0), T^\pm$ | $\tilde{W}'^0, \tilde{W}'^\pm$ | $(\mathbf{1},\mathbf{3},0)$ |
| $\mathbf{S}$ | $S = \frac{1}{\sqrt{2}}(S_R + iS_I)$ | $\tilde{B}'^0$ | $(\mathbf{1},\mathbf{1},0)$ |

Table 1: Field content in Dirac gaugino models, apart from quark and lepton superfields, and possible R-symmetry charges prior to the addition of the *explicit* R-symmetry breaking term $B_\mu$; note that $R_H$ is arbitrary. Top panel: chiral and gauge multiplet fields as in the MSSM; bottom panel: chiral and gauge multiplet fields added to those of the MSSM to allow Dirac masses for the gauginos.

Dirac fermions. For a concise review of the EW sector of the MRSSM see [50] section 2.3; in appendix A.1 we review the EW sector of that model to contrast with the MDGSSM, with some additional comments about R-symmetry breaking and its relevance to the phenomenology that we discuss later. However, in that class of models the phenomenology is different to that described here.

The second major class of models is those for which the R-symmetry is violated. This includes the minimal choices in terms of numbers of additional fields – the SOHDM [28], "MSSM without $\mu$ term" [81] and MDGSSM, as well as extensions with more fields, e.g. to allow unification of the gauge couplings, such as the CMDGSSM [72, 77]. The constraints on the EW-ino sectors of these models should be broadly similar. Crucially in these models – in contrast to those where the EW-inos are exactly Dirac – the neutralinos are pseudo-Dirac Majorana fermions. This means that they come in pairs with a small mass splitting, in particular between the neutral partner of a bino or wino LSP and the LSP itself. This has significant consequences for dark matter in the model, as has already been explored in e.g. [75, 77]: coannihilation occurs naturally. However, we shall also see here that it has significant consequences for the collider constraints: the decays from $\tilde{\chi}_2^0$ to $\tilde{\chi}_1^0$ are generally soft and hard to observe, and lead to a long-lived particle in some of the parameter space.

## 2.2 Electroweakinos in the MDGSSM

Here we shall summarise the important features of the EW-ino sector of the MDGSSM. Our notation and definitions are essentially identical to [75], to which we refer the reader for a more complete treatment.

The MDGSSM can be defined as the minimal extension of the MSSM allowing for Dirac gaugino masses. We add one adjoint chiral superfield for each gauge group, and nothing else: the field content is summarised in Table 1. We also assume that there is an under-

lying R-symmetry that prevents R-symmetry-violating couplings in the superpotential and supersymmetry-breaking sector, *except* for an explicit breaking in the Higgs sector through a (small) $B_\mu$ term. This was suggested in the "MSSM without $\mu$-term" [81] as such a term naturally has a special origin through gravity mediation; it is also stable under renormalisation group evolution, as the $B_\mu$ term does not induce other R-symmetry violating terms.

The singlet and triplet fields can have new superpotential couplings with the Higgs,

$$W = W_{\mathrm{MSSM}} + \lambda_S \mathbf{S}\, \mathbf{H_u} \cdot \mathbf{H_d} + 2\lambda_T\, \mathbf{H_d} \cdot \mathbf{T H_u}\,. \tag{2}$$

These new couplings may or may not have an underlying motivation from $N = 2$ supersymmetry, which has been explored in detail [59]. After electroweak symmetry breaking (EWSB), we obtain 6 neutralino and 3 chargino mass eigenstates (as compared to 4 and 2, respectively, in the MSSM). The neutralino mass matrix $\mathcal{M}_N$ in the basis $(\tilde{B}', \tilde{B}, \tilde{W}'^0, \tilde{W}^0, \tilde{H}_d^0, \tilde{H}_u^0)$ is given by

$$\mathcal{M}_N = \tag{3}$$

$$\begin{pmatrix} 0 & m_{DY} & 0 & 0 & -\frac{\sqrt{2}\lambda_S}{g_Y}m_Z s_W s_\beta & -\frac{\sqrt{2}\lambda_S}{g_Y}m_Z s_W c_\beta \\ m_{DY} & 0 & 0 & 0 & -m_Z s_W c_\beta & m_Z s_W s_\beta \\ 0 & 0 & 0 & m_{D2} & -\frac{\sqrt{2}\lambda_T}{g_2}m_Z c_W s_\beta & -\frac{\sqrt{2}\lambda_T}{g_2}m_Z c_W c_\beta \\ 0 & 0 & m_{D2} & 0 & m_Z c_W c_\beta & -m_Z c_W s_\beta \\ -\frac{\sqrt{2}\lambda_S}{g_Y}m_Z s_W s_\beta & -m_Z s_W c_\beta & -\frac{\sqrt{2}\lambda_T}{g_2}m_Z c_W s_\beta & m_Z c_W c_\beta & 0 & -\mu \\ -\frac{\sqrt{2}\lambda_S}{g_Y}m_Z s_W c_\beta & m_Z s_W s_\beta & -\frac{\sqrt{2}\lambda_T}{g_2}m_Z c_W c_\beta & -m_Z c_W s_\beta & -\mu & 0 \end{pmatrix},$$

where $s_W = \sin\theta_W$, $s_\beta = \sin\beta$ and $c_\beta = \cos\beta$; $\tan\beta = v_u/v_d$ is the ratio of the Higgs vevs; $m_{DY}$ and $m_{D2}$ are the 'bino' and 'wino' Dirac mass parameters; $\mu$ is the higgsino mass term, and $\lambda_S$ and $\lambda_T$ are the couplings between the singlet and triplet fermions with the Higgs and higgsino fields. By diagonalising eq. (3), one obtains pairs of bino-like, wino-like and higgsino-like neutralinos,[1] with small mass splittings *within* the bino or wino pairs induced by $\lambda_S$ or $\lambda_T$, respectively. For instance, if $m_{DY}$ is sufficiently smaller than $m_{D2}$ and $\mu$, we find mostly bino/U(1) adjoint $\tilde{\chi}_{1,2}^0$ as the lightest states with a mass splitting given by

$$m_{\tilde{\chi}_2^0} - m_{\tilde{\chi}_1^0} \simeq \left| 2\frac{M_Z^2 s_W^2}{\mu}\frac{(2\lambda_S^2 - g_Y^2)}{g_Y^2}c_\beta s_\beta \right|\,. \tag{4}$$

Alternative approximate formulae for the mass-splitting in other cases were also given in [75].

Turning to the charged EW-inos, the chargino mass matrix in the basis $v^+ = (\tilde{W}'^+, \tilde{W}^+, \tilde{H}_u^+)$, $v^- = (\tilde{W}'^-, \tilde{W}^-, \tilde{H}_d^-)$ is given by:

$$\mathcal{M}_C = \begin{pmatrix} 0 & m_{D2} & \frac{2\lambda_T}{g_2}m_W c_\beta \\ m_{D2} & 0 & \sqrt{2}m_W s_\beta \\ -\frac{2\lambda_T}{g_2}m_W s_\beta & \sqrt{2}m_W c_\beta & \mu \end{pmatrix}, \tag{5}$$

This can give a higgsino-like $\tilde{\chi}^\pm$ as in the MSSM, but we now have *two* wino-like $\tilde{\chi}^\pm$ – the latter ones again with a small splitting driven by $\lambda_T$. A wino LSP therefore consists of a set of *two* neutral Majorana fermions and *two* Dirac charginos, all with similar masses.

---

[1]For simplicity, we refer to the mostly bino/U(1) adjoint states collectively as binos, and to the mostly wino/SU(2) adjoint ones as winos.

Note that in both eqs. (3) and (5), Majorana mass terms are absent, since we assume that the only source of R-symmetry breaking in the model is the $B_\mu$ term. If we were to add Majorana masses for the gauginos, or supersymmetric masses for the singlet/triplet fields, then they would appear as diagonal terms in the above matrices (see e.g. [75] for the neutralino and chargino mass matrices with such terms included), and would generically lead to larger splitting of the pseudo-Dirac states.

# 3 Setup of the numerical analysis

## 3.1 Parameter scan

We now turn to the numerical analysis. Focusing solely on the EW-ino sector, the parameter space we consider is:

$$0 < m_{DY}, \, m_{D2}, \, \mu < 2 \text{ TeV}; \quad 1.7 < \tan\beta < 60; \quad -3 < \lambda_S, \, \lambda_T < 3. \tag{6}$$

The rest of the sparticle content of the MDGSSM is assumed to be heavy, with slepton masses fixed at 2 TeV, soft masses of the 1st/2nd and 3rd generation squarks set to 3 TeV and 3.5 TeV, respectively, and gluino masses set to 4 TeV. The rest of parameters are set to the same values as in [69]; in particular trilinear $A$-terms are set to zero.

The mass spectrum and branching ratios are computed with SPheno v4.0.3 [82, 83], using the DiracGauginos model [84] exported from SARAH [85–88]. This is interfaced to micrOMEGAs v5.2 [89–91][2] for the computation of the relic density, direct detection limits and other constraints explained below. To efficiently scan over the EW-ino parameters, eq. (6), we implemented a Markov Chain Monte Carlo (MCMC) Metropolis-Hastings algorithm that walks towards the minimum of the negative log-likelihood function, $-\log(L)$, defined as

$$-\log(L) = \chi^2_{\Omega h^2} - \log(p_{\text{X1T}}) + \log(m_{\text{LSP}}). \tag{7}$$

Here,

- $\chi^2_{\Omega h^2}$ is the $\chi^2$-test of the computed neutralino relic density compared to the observed relic density, $\Omega h^2_{\text{Planck}} = 0.12$ [92]. In a first scan, this is implemented as an upper bound only, that is

$$\chi^2_{\Omega h^2} = \frac{(\Omega h^2 - \Omega h^2_{\text{Planck}})^2}{\Delta^2_\Omega} \tag{8}$$

  if $\Omega h^2 > \Omega h^2_{\text{Planck}}$, and zero otherwise. In a second scan, eq. (8) is applied as a two-sided bound for all $\Omega h^2$. Allowing for a 10% theoretical uncertainty (as a rough estimate, to account e.g. for the fact that the relic density calculation is done at the tree level only), we take $\Delta^2_\Omega = 0.1 \, \Omega h^2_{\text{Planck}}$.

- $p_{\text{X1T}}$ is the $p$-value for the parameter point being excluded by XENON1T results [93]. The confidence level (CL) being given by $1 - p_{\text{X1T}}$, a value of $p_{\text{X1T}} = 0.1 \, (0.05)$ corresponds to 90% (95%) CL exclusion. To compute $p_{\text{X1T}}$, the LSP-nucleon scattering cross sections are rescaled by a factor $\Omega h^2 / \Omega h^2_{\text{Planck}}$.

---

[2]More precisely, we used a private pre-release version of micrOMEGAs v5.2, which does however give the same results as the official release.

- $m_{\text{LSP}}$ is the mass of the neutralino LSP, added to avoid the potential curse of dimensionality.[3]

In order to explore the whole parameter space, a small jump probability is introduced which prevents the scan from getting stuck in local minima of $-\log(L)$. We ran several Markov Chains from different, randomly drawn starting points; the algorithm is outlined step-by-step in Appendix A.2.

The light Higgs mass, $m_h$, also depends on the input parameters, and it is thus important to find the subset of the parameter space where it agrees with the experimentally measured value. Instead of including $m_h$ in the likelihood function, eq. (7), that guides the MCMC scan, we implemented a Random Forest Classifier that predicts whether a given input point has $m_h$ within a specific target range. As the desired range we take $120 < m_h < 130$ GeV, assuming $m_h \simeq 125$ GeV can then always be achieved by tuning parameters in the stop sector. Points outside $120 < m_h < 130$ GeV are discarded. This significantly speeds up the scan. Details on the Higgs mass classifier are given in Appendix A.3.

In the various MCMC runs we kept for further analysis all points scanned over, which

1. have a neutralino LSP (charged LSPs are discarded);

2. have a light Higgs boson in the range $120 < m_h < 130$ GeV (see above);

3. avoid mass limits from supersymmetry searches at LEP as well as constraints from the $Z$ boson invisible decay width as implemented in microOMEGAs [90];

4. have $\Omega h^2 < 1.1 \, \Omega h^2_{\text{Planck}}$ (or $\Omega h^2 = \Omega h^2_{\text{Planck}} \pm 10\%$) and

5. have $p_{\text{X1T}} > 0.1$.

With the procedure outlined above, many points with very light LSP, in the mass range below $m_h/2$ and even below $m_Z/2$, are retained. We therefore added two more constraints *a posteriori*. Namely, we require for valid points that

6. $\Delta\rho$ lies within $3\sigma$ of the measured value $\Delta\rho_{\text{exp}} = (3.9 \pm 1.9) \times 10^{-4}$ [94], the $3\sigma$ range being chosen in order to include the SM value of $\Delta\rho = 0$;

7. signal-strength constraints from the SM-like Higgs boson as computed with Lilith-2 [95] give a $p$-value of $p_{\text{Lilith}} > 0.05$; this eliminates in particular points in which $m_{\text{LSP}} < m_h/2$, where the branching ratio of the SM-like Higgs boson into neutralinos or charginos is too large.

Points which do not fulfil these conditions are discarded. We thus collect in total 52550 scan points (out of $\mathcal{O}(10^6)$ tested points), which fulfil all constraints, as the basis for our phenomenological analysis.

## 3.2 Treatment of electroweakino decays

As argued above and will become apparent in the next section, many of the interesting scenarios in the MDGSSM feature the second neutralino and/or the lightest chargino very close in mass

---

[3]Due to the exponential increase in the volume of the parameter space, one risks having too many points with an $m_{\text{LSP}}$ at the TeV scale. Current LHC searches are not sensitive to such heavy EW-inos.

to the LSP. With mass splittings of $\mathcal{O}(1)$ GeV, $\tilde{\chi}_1^\pm$ or $\tilde{\chi}_2^0$ decays into $\tilde{\chi}_1^0+$ pion(s) and $\tilde{\chi}_2^{0,\pm}$ decays into $\tilde{\chi}_1^{0,\pm} + \gamma$ become important. These decays were in the first case not implemented, and in the second not treated correctly in the standard SPheno/SARAH. We therefore describe below how these decays are computed in our analysis; the corresponding modified code is available online [96].[4]

Note that the precise calculation of the chargino and neutralino decays is important not only for the collider signatures (influencing branching ratios and decay lengths), but can also impact the DM relic abundance and/or direct detection cross sections.

### 3.2.1 Chargino decays into pions

When the mass splitting between chargino and lightest neutralino becomes sufficiently small, three-body decays via an off-shell $W$-boson, $\tilde{\chi}_1^\pm \to \tilde{\chi}_1^0 + (W_\mu^\pm)^*$ start to dominate. However, as pointed out in e.g. Appendix A of [98] (see also [100] and references therein), when $\Delta m \lesssim 1.5$ GeV it is not accurate to describe the $W^*$ decays in terms of quarks, but instead we should treat the final states as one, two or three pions (with Kaon final states being Cabibbo-suppressed)[5]; and for $\Delta m < m_\pi$ the hadronic channel is closed. Surprisingly, these decays have not previously been fully implemented in spectrum generators; SPheno contains only decays to single pions from neutralinos or charginos in the MSSM via an off-shell W or Z boson, and SARAH does not currently include even these. A full generic calculation of decays with mesons as final states for both charged and neutral EW-inos (and its implementation in SARAH) should be presented elsewhere; for this work we have adapted the results of [97–99] which include only the decay via an off-shell W:

$$\Gamma(\tilde{\chi}_1^- \to \tilde{\chi}_1^0 \pi^-) = \frac{f_\pi^2 G_F^2}{2\pi g_2^2} \frac{|\vec{k}_\pi|}{\widetilde{m}_-^2} \left\{ \left(|c_L|^2 + |c_R|^2\right)\left[\left(\widetilde{m}_-^2 - \widetilde{m}_0^2\right)^2 - m_\pi^2\left(\widetilde{m}_-^2 + \widetilde{m}_0^2\right)\right] \right.$$
$$\left. + 4\widetilde{m}_0\widetilde{m}_- m_\pi^2 \text{Re}\left(c_L c_R^*\right) \right\} \tag{9}$$

$$\Gamma(\tilde{\chi}_1^- \to \tilde{\chi}_1^0 \pi^- \pi^0) = \frac{G_F^2}{192\pi^3 g_2^2 \widetilde{m}_-^3} \int_{4m_\pi^2}^{(\Delta m_{\tilde{\chi}_1})^2} dq^2 \left|F(q^2)\right|^2 \left(1 - \frac{4m_\pi^2}{q^2}\right)^{3/2} \lambda^{1/2}(\widetilde{m}_-^2, \widetilde{m}_0^2, q^2)$$
$$\left\{\left[|c_L|^2 + |c_R|^2\right]\left[q^2\left(\widetilde{m}_-^2 + \widetilde{m}_0^2 - 2q^2\right) + \left(\widetilde{m}_-^2 - \widetilde{m}_0^2\right)^2\right] - 12\text{Re}(c_L c_R^*)q^2\widetilde{m}_-\widetilde{m}_0\right\}; \tag{10}$$

$$\Gamma(\tilde{\chi}_1^- \to \tilde{\chi}_1^0 3\pi) = \frac{G_F^2}{6912\pi^5 g_2^2 \widetilde{m}_-^3 f_\pi^2} \int_{9m_\pi^2}^{(\Delta m_{\tilde{\chi}_1})^2} dq^2 \lambda^{1/2}(\widetilde{m}_-^2, \widetilde{m}_0^2, q^2)\left|BW_a(q^2)\right|^2 g(q^2)$$
$$\left\{\left[|c_L|^2 + |c_R|^2\right]\left[\widetilde{m}_-^2 + \widetilde{m}_0^2 - 2q^2 + \frac{\left(\widetilde{m}_-^2 - \widetilde{m}_0^2\right)^2}{q^2}\right] - 12\text{Re}(c_L c_R^*)\widetilde{m}_-\widetilde{m}_0\right\}. \tag{11}$$

Here $\widetilde{m}_-, \widetilde{m}_0$ are the masses of the $\tilde{\chi}_1^-, \tilde{\chi}_1^0$ respectively, $\vec{k}_\pi = \lambda^{1/2}(\widetilde{m}_-^2, \widetilde{m}_0^2, m_\pi^2)/(2\widetilde{m}_-)$ is the pion's 3-momentum in the chargino rest frame, and $f_\pi \simeq 93$ MeV is the pion decay constant. The couplings $c_L, c_R$ are the left and right couplings of the chargino and neutralino to the W-boson, which can be defined as $\mathcal{L} \supset -\overline{\tilde{\chi}_1^-}\gamma^\mu(c_L P_L + c_R P_R)\chi_0 W_\mu^-$. The couplings of the W-boson

---

[4]We leave the decays $\tilde{\chi}_i^0$ to $\tilde{\chi}_j^\pm+$ pion(s) to future work.

[5]As the mass difference is raised above $\Delta m = 1.5$ GeV is it found numerically that, with many hadronic decay modes being kinematically open, there is a smooth transition to a description in terms of quarks.

Large $|\mu|$ limiting case, MSSM

Figure 1: Chargino decays in the MSSM limit of our model; see text for details.

to the light quarks and the W mass are encoded in $G_F$; in `SARAH` we make the substitution $G_F^2 \rightarrow g_2^2 |c_L^{u\bar{d}W}|^2/(16M_W^4)$, where $c_L^{u\bar{d}W}$ is the coupling of the up and down quarks to the W-boson.

While the single pion decay can be simply understood in terms of the overlap of the axial current with the pion, the two- and three-pion decays proceed via exchange of virtual mesons which then decay to pions. The form factors for these processes are then determined by QCD, and so working at leading order in the electroweak couplings we can use experimental data for processes involving the same final states; in this case we can use $\tau$ lepton decays. The two-pion decays are dominated by $\rho$ and $\rho'$ meson exchange, and the form factor $F(q^2)$ was defined in eqs. (A3) and (A4) of [98]. The expressions for the Breit–Wigner propagator $BW_a$ of the $a_1$ meson (and *not* the $a_2$ meson as stated in [97–99]), which dominates $3\pi$ production, as well as for the three-pion phase space factor $g(q^2)$ can be found in eqs. (3.16)–(3.18) of [100]. As in [97–99] we use the propagator without "dispersive correction," and so include a factor of 1.35 to compensate for the underestimate of $\tau^- \rightarrow 3\pi\nu_\tau$ decays by 35%. Note finally that the three-pion decay includes both $\pi^-\pi^0\pi^0$ and $\pi^-\pi^-\pi^+$ modes, which are assumed to be equal.

For comparison with [97–99], in Figure 1 we reproduce Fig. 6 from [98] (same as Fig. 1 in [99]) with our code by taking the MSSM-limit of our model; we add Majorana gaugino masses for the the wino fixed at $M_2 = 200$ GeV and scan over values for the bino mass of $M_1 \in [210, 220]$ GeV while taking $\mu = 2000$ GeV and adding supersymmetric masses for the **S** and **T** fields of $M_S = M_T = 1$ TeV. Keeping $\tan\beta = 34.664$ and $B_\mu = (1\,\text{TeV})^2$ we have a spectrum with effectively only Majorana charginos and neutralinos, which can be easily tuned in mass relative to each other by changing the bino mass.

In Figure 2 we show the equivalent expressions in the case of interest for this paper, where there are no Majorana masses for the gauginos. We take $\tan\beta = 34.664, \mu = 2$ TeV, $v_T = -0.568$ GeV, $v_S = 0.92$ GeV, $\lambda_S = -0.2, \sqrt{2}\lambda_T = 0.2687, m_{D2} = 200$ GeV, and vary $m_{DY}$ between 210 and 221 GeV. We find identical behaviour for both models, except the overall

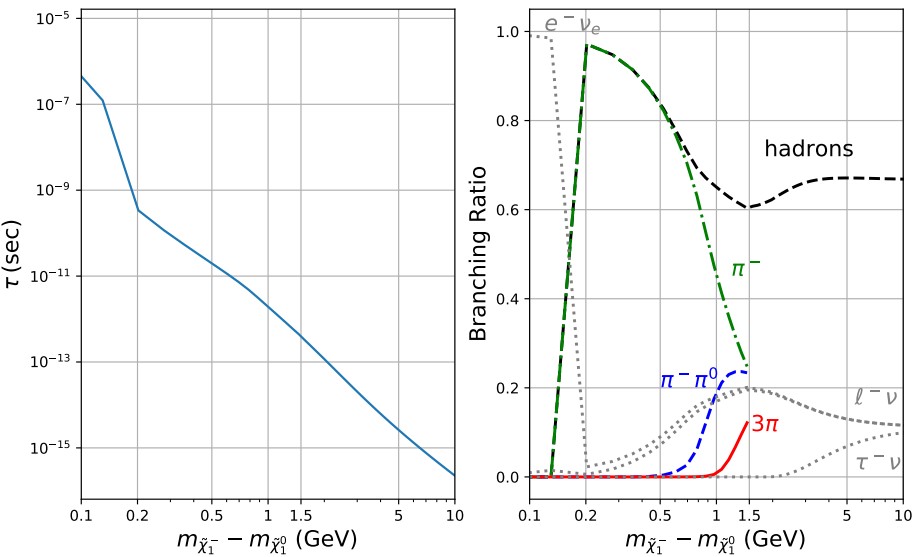

Figure 2: Chargino decays in the MDGSSM.

decay rate is slightly different; and note that in this scenario we have $\tilde{\chi}_2^0$ almost degenerate with $\tilde{\chi}_1^0$, so we include decays of $\tilde{\chi}_1^{\pm}$ to *both* states of the pseudo-Dirac LSP.

Finally, we implemented the decays of neutralinos to single pions via the expression

$$\Gamma(\tilde{\chi}_2^0 \to \tilde{\chi}_1^0 \pi^0) = \frac{f_\pi^2 G_F^2 c_W^2}{2\pi g_2^2} \frac{|\vec{k}_\pi|}{\widetilde{m}_2^2} \left\{ \left(|c_L|^2 + |c_R|^2\right)\left[\left(\widetilde{m}_2^2 - \widetilde{m}_1^2\right)^2 - m_\pi^2\left(\widetilde{m}_2^2 + \widetilde{m}_1^2\right)\right]\right.$$
$$\left. + 4\widetilde{m}_1\widetilde{m}_2 m_\pi^2 \mathrm{Re}\left(c_L c_R^*\right) \right\} \tag{12}$$

where now $\widetilde{m}_{1,2}$ are the masses of $\tilde{\chi}_{1,2}^0$ and $c_L, c_R$ are the couplings for the neutralinos to the $Z$-boson analogously defined as above; since the neutralino is Majorana in nature we must have $c_R = -c_L^*$.

### 3.2.2 Neutralino decays into photons

In the MDGSSM, the mass splitting between the two lightest neutralinos is naturally small.[6] Therefore in a significant part of the parameter space the dominant $\tilde{\chi}_2^0$ decay mode is the loop-induced process $\tilde{\chi}_2^0 \to \tilde{\chi}_1^0 + \gamma$. This is controlled by an effective operator

$$\mathcal{L} = \overline{\Psi}_1 \gamma^\mu \gamma^\nu (C_{12} P_L + C_{12}^* P_R) \Psi_2 F_{\mu\nu}, \tag{13}$$

where $\Psi_i \equiv \begin{pmatrix} \chi_i^0 \\ \overline{\chi}_i^0 \end{pmatrix}$ is a Majorana spinor, and yields

$$\Gamma(\tilde{\chi}_2^0 \to \tilde{\chi}_1^0 + \gamma) = \frac{|C_{12}|^2}{2\pi} \frac{(m_{\tilde{\chi}_2}^2 - m_{\tilde{\chi}_1}^2)^3}{m_{\tilde{\chi}_2}^3}. \tag{14}$$

---

[6]This could be even more so in the case of the MRSSM with a small R-symmetry violation.

Our expectation (and indeed as we find for most of our points) is that $|C_{12}| \sim 10^{-5}$–$10^{-6}\,\mathrm{GeV}^{-1}$.

This loop decay process is calculated in SPheno/SARAH using the routines described in [101]. However, we found that the handling of fermionic two-body decays involving photons or gluons was not correctly handled in the spin structure summation. Suppose we have S-matrix elements $\mathcal{M}$ for a decay $F(p_1) \to F(p_2) + V(p_3)$ with a vector having wavefunction $\varepsilon_\mu$, then we can decompose the amplitudes according to their Lorentz structures (putting $v_i$ for the antifermion wavefunctions) as

$$\mathcal{M} = \varepsilon_\mu \mathcal{M}^\mu = \varepsilon_\mu(p_3)\left[ x_1 \bar{v}_1 P_L \gamma^\mu v_2 + x_2 \bar{v}_1 P_R \gamma^\mu v_2 + p_1^\mu x_3 \bar{v}_1 P_L v_2 + p_1^\mu x_4 \bar{v}_1 P_R v_2 \right]. \tag{15}$$

This is the decomposition made in SARAH which computes the values of the amplitudes $\{x_i\}$. Now, if $V$ is massless, and since $\mathcal{M}$ is an S-matrix element, the Ward identity requires $(p_3)_\mu \mathcal{M}^\mu = 0$ (note that this requires that we include self-energy diagrams in the case of charged fermions), and this leads to two equations relating the $\{x_i\}$:

$$x_3 = \frac{m_1 x_2 - m_2 x_1}{p_1 \cdot p_3}, \qquad x_4 = \frac{m_1 x_1 - m_2 x_2}{p_1 \cdot p_3}, \qquad \text{where } p_1 \cdot p_3 = \frac{1}{2}(m_1^2 - m_2^2). \tag{16}$$

Here, $m_1$ and $m_2$ are the masses of the first and second fermion, respectively. Performing the spin and polarisation sums naively, we have the matrix

$$\sum_{\text{spins, polarisations}} \mathcal{M}\mathcal{M}^* \equiv x_i \mathcal{M}_{ij} x_j^*, \tag{17}$$

$$\mathcal{M}_{ij} = \begin{pmatrix} 2(m_1^2 + m_2^2) & -8m_1 m_2 & 2m_1^2 m_2 & m_1(m_1^2 + m_2^2) \\ -8m_1 m_2 & 2(m_1^2 + m_2^2) & m_1(m_1^2 + m_2^2) & 2m_1^2 m_2 \\ 2m_1^2 m_2 & m_1(m_1^2 + m_2^2) & -m_1^2(m_1^2 + m_2^2) & -2m_1^3 m_2 \\ m_1(m_1^2 + m_2^2) & 2m_1^2 m_2 & -2m_1^3 m_2 & -m_1^2(m_1^2 + m_2^2) \end{pmatrix}.$$

When we substitute in the Ward identities and re-express as just $x_1, x_2$ we have

$$\sum_{\text{spins, polarisations}} \mathcal{M}\mathcal{M}^* = (x_1, x_2) \begin{pmatrix} 2(m_1^2 + m_2^2) & -4m_1 m_2 \\ -4m_1 m_2 & 2(m_1^2 + m_2^2) \end{pmatrix} \begin{pmatrix} x_1^* \\ x_2^* \end{pmatrix}. \tag{18}$$

This matrix will yield real, positive-definite widths for any value of the matrix elements $x_1, x_2$, whereas this is not manifestly true for eq. (17). Therefore as of SARAH version 4.14.3 we implemented the spin summation for loop decay matrix elements given in eq. (18), i.e. in such decays we compute the Lorentz structures corresponding to $x_1, x_2$ and ignore $x_3, x_4$.

This applies to all $\tilde{\chi}^0_{i \neq 1} \to \tilde{\chi}^0_1 \gamma$ and $\tilde{\chi}^\pm_{j \neq 1} \to \tilde{\chi}^\pm_1 \gamma$ transitions.

# 4 Results

## 4.1 Properties of viable scan points

We are now in the position to discuss the results from the MCMC scans. We begin by considering the properties of the $\tilde{\chi}^0_1$ as a DM candidate. Figure 3(a) shows the bino, wino and higgsino composition of the $\tilde{\chi}^0_1$ when only an upper bound on $\Omega h^2$ is imposed; all points in the plot also satisfy XENON1T ($p_{\text{X1T}} > 0.1$) and all other constraints listed in section 3.1. We

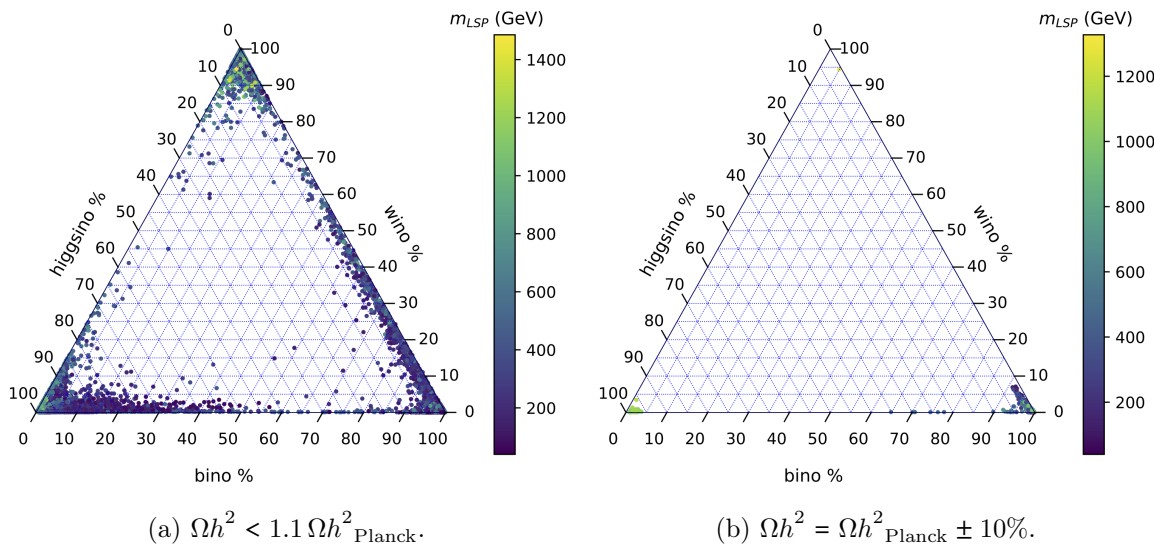

(a) $\Omega h^2 < 1.1 \, \Omega h^2{}_{\text{Planck}}$.          (b) $\Omega h^2 = \Omega h^2{}_{\text{Planck}} \pm 10\%$.

Figure 3: Bino, wino and higgsino admixtures of the LSP in the region where it makes up for (a) at least a part or (b) all of the DM abundance; limits from XENON1T and all other constraints listed in section 3.1 are also satisfied. The colour denotes the mass of the LSP.

see that cases where the $\tilde{\chi}_1^0$ is a mixture of all states (bino, wino and higgsino) are excluded, while cases where it is a mixture of only two states, with one component being dominant, can satisfy all constraints. Also noteworthy is that there are plenty of points in the low-mass region, $m_{\text{LSP}} < 400$ GeV.

Figure 3(b) shows the points where the $\tilde{\chi}_1^0$ makes for all the DM abundance. This, of course, imposes much stronger constraints. In general, scenarios with strong admixtures of two or more EW-ino states are excluded and the valid points are confined to the corners of (almost) pure bino, wino or higgsino. Similar to the MSSM, the higgsino and especially the wino DM cases are heavy, with masses $\gtrsim 1$ TeV, and only about a 5% admixture of another interaction eigenstate; in the wino case, the MCMC scan gave only one surviving point within the parameter ranges scanned over. Light masses are found only for bino-like DM; in this case there can also be slightly larger admixtures of another state: concretely we find up to about 10% wino or up to 35% higgsino components.

As mentioned, we assume that all other sparticles besides the EW-inos are heavy. Hence, co-annihilations of EW-inos which are close in mass to the LSP must be the dominating processes to achieve $\Omega h^2$ of the order of 0.1 or below. The relation between mass, bino/wino/higgsino nature of the LSP, relic density and mass difference to the next-to-lightest sparticle (NLSP) is illustrated in Figure 4. The three panels of this figure show $m_{\text{LSP}}$ vs. $\Omega h^2$ for the points from Figure 3(a), where the LSP is > 50% bino, wino, or higgsino, respectively. The NLSP–LSP mass difference is shown in colour, while different symbols denote neutral and charged NLSPs. Two things are apparent besides the dependence of $\Omega h^2$ on $m_{\tilde{\chi}_1^0}$ for the different scenarios:

1. All three cases feature small NLSP–LSP mass differences. For a wino-like LSP, this mass difference is at most 3 GeV. For bino-like and higgsino-like LSPs it can go up to nearly 25 GeV, though for most points it is just few GeV.

2. The NLSP can be neutral or charged, that is in all three cases we can have mass orderings

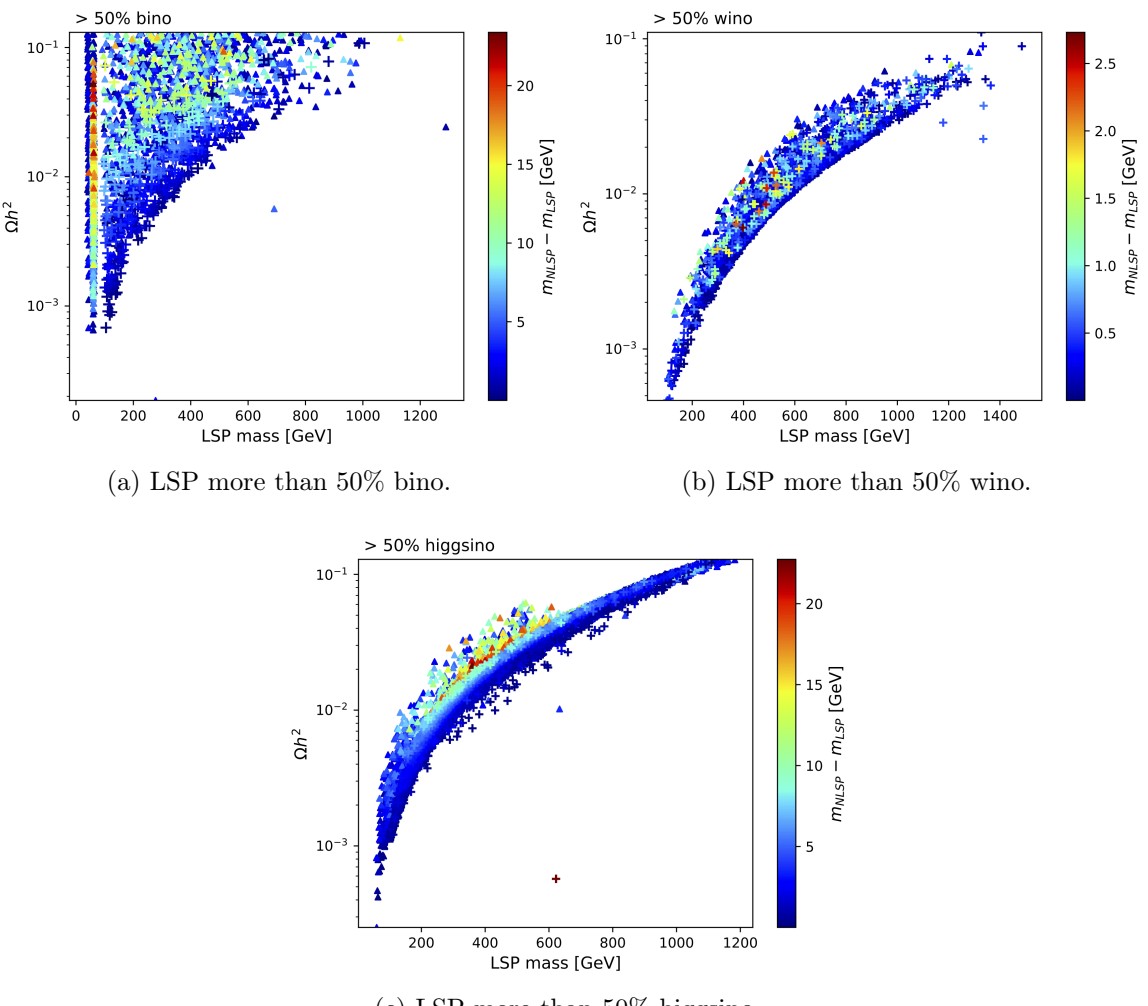

(a) LSP more than 50% bino.

(b) LSP more than 50% wino.

(c) LSP more than 50% higgsino.

Figure 4: $m_{\text{LSP}}$ vs. $\Omega h^2$ for points from Figure 3(a), where (a) LSP > 50% bino, (b) LSP > 50% wino, and (c) LSP > 50% higgsino. In color, the NLSP–LSP mass difference. Triangles represent neutral NLSPs while crosses represent charged NLSPs.

$$\tilde{\chi}_1^0 < \tilde{\chi}_1^\pm < \tilde{\chi}_2^0 \text{ as well as } \tilde{\chi}_1^0 < \tilde{\chi}_2^0 < \tilde{\chi}_1^\pm.$$

For bino-like LSP points outside the $Z$ and Higgs-funnel regions, a small mass difference between the LSP and NLSP is however not sufficient—co-annihilations with other nearby states are required to achieve $\Omega h^2 \lesssim 0.132$. Indeed, as shown in Figure 5, we have $m_{D2} \approx m_{DY}$, with typically $m_{D2}/m_{DY} \approx 0.9$–1.4, over much of the bino-LSP parameter space outside the funnel regions. This leads to bino-wino co-annihilation scenarios like also found in the MSSM. The scattered points with large ratios $m_{D2}/m_{DY}$ have $\mu \approx m_{DY}$, i.e. a triplet of higgsinos close to the binos. Outside the funnel regions, the bino-like LSP points therefore feature $m_{\tilde{\chi}_1^\pm} - m_{\tilde{\chi}_1^0} \lesssim 30$ GeV and $m_{\tilde{\chi}_{3,4}^0} - m_{\tilde{\chi}_1^0} \lesssim 60$ GeV in addition to $m_{\tilde{\chi}_2^0} - m_{\tilde{\chi}_1^0} \lesssim 20$ GeV.

For completeness we also give the maximal mass differences found within triplets (quadruplets) of higgsino (wino) states in the higgsino (wino) LSP scenarios. Concretely we have $m_{\tilde{\chi}_2^0} - m_{\tilde{\chi}_1^0} \lesssim 15$ GeV and $m_{\tilde{\chi}_1^\pm} - m_{\tilde{\chi}_1^0} \lesssim 50$–10 GeV (decreasing with increasing $m_{\tilde{\chi}_1^0}$) in the

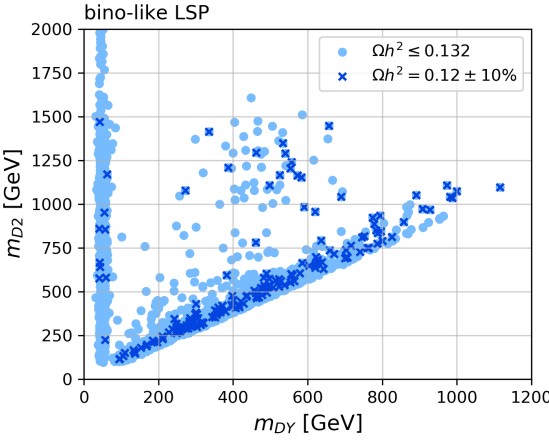

Figure 5: $m_{DY}$ vs. $m_{D2}$ for scan points with a bino-like LSP, cf. Figure 4(a).

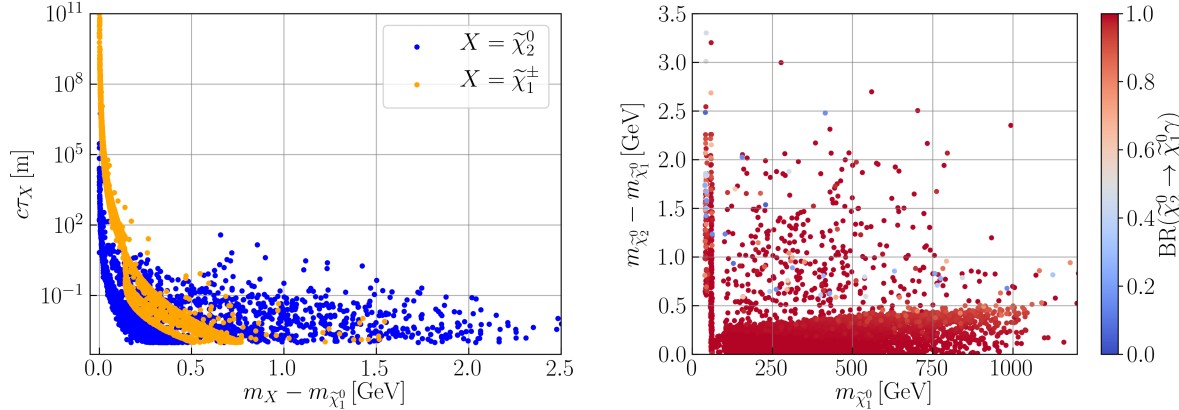

Figure 6: Left: Mean decay length $c\tau$ as a function of the mass difference with the LSP, for all points with long-lived particles ($c\tau > 1$ mm); blue points have a neutralino and orange points a chargino LLP. Right: $m_{\tilde{\chi}_1^0}$ vs. $m_{\tilde{\chi}_2^0} - m_{\tilde{\chi}_1^0}$ for points with long-lived neutralinos; the branching ratio of the loop decay $\tilde{\chi}_2^0 \to \tilde{\chi}_1^0 \gamma$ is indicated in colour.

higgsino LSP case. In the wino LSP case, $m_{\tilde{\chi}_1^\pm} - m_{\tilde{\chi}_1^0} \lesssim 4$ GeV, while $m_{\tilde{\chi}_2^0, \tilde{\chi}_2^\pm} - m_{\tilde{\chi}_1^0} \lesssim 20$ GeV (though mostly below 10 GeV). However, as noted before, either mass ordering, $m_{\tilde{\chi}_2^0} < m_{\tilde{\chi}_1^\pm}$ or $m_{\tilde{\chi}_1^\pm} < m_{\tilde{\chi}_2^0}$ is possible.

An important point to note is that the mass differences are often so small that the NLSP (and sometimes even the NNLSP) becomes long-lived on collider scales, i.e. it has a potentially visible decay length of $c\tau > 1$ mm. This is illustrated in Figure 6, which shows in the left panel the mean decay length of the LLPs as function of their mass difference to the LSP. Long-lived charginos will lead to charged tracks in the detector, while long-lived neutralinos could potentially lead to displaced vertices. However, given the small mass differences involved, the decay products of the latter will be very soft. The right panel in Figure 6 shows the importance of the radiative decay of long-lived $\tilde{\chi}_2^0$s in the plane of $\tilde{\chi}_1^0$ mass vs. $\tilde{\chi}_2^0$–$\tilde{\chi}_1^0$ mass difference. As can be seen, decays into (soft) photons are clearly dominant.

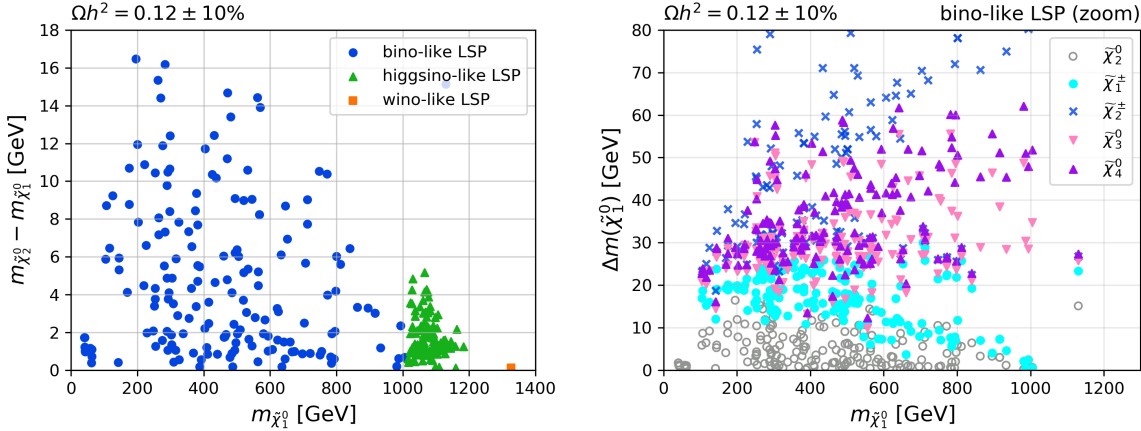

Figure 7: Left: $m_{\rm LSP}$ vs. NLSP–LSP mass difference for points from Figure 3(b); points with bino-, higgsino-, and wino-like LSP are shown in blue, green and orange, respectively. Right: mass differences $\Delta m$ of $\tilde{\chi}^0_{2,3,4}$ and $\tilde{\chi}^\pm_{1,2}$ to the $\tilde{\chi}^0_1$ as function of the $\tilde{\chi}^0_1$ mass, for the bino DM points of the right panel.

Let us now turn to the region where the $\tilde{\chi}^0_1$ would account for all the DM. Figure 7 (left) shows the points with $\Omega h^2 = \Omega h^2_{\rm Planck} \pm 10\%$ in the plane of $m_{\tilde{\chi}^0_1}$ vs. $m_{\tilde{\chi}^0_2} - m_{\tilde{\chi}^0_1}$. Points with bino-like, higgsino-like and wino-like $\tilde{\chi}^0_1$ are distinguished by different colours and symbols. As expected from the discussion above, there are three distinct regions of bino-like, higgsino-like and wino-like DM, indicated in blue, green and orange, respectively.

From the collider point of view, the bino-like DM region is perhaps the most interesting one, as it has masses below a TeV. We find that, in this case, the NLSP is always the $\tilde{\chi}^0_2$ with mass differences $m_{\tilde{\chi}^0_2} - m_{\tilde{\chi}^0_1}$ ranging from about 0.2 GeV to 16 GeV. As already pointed in [75, 76], this small mass splitting helps achieve the correct relic density through $\tilde{\chi}^0_{1,2}$ co-annihilation. In the region of $m_{\tilde{\chi}^0_1} = 100 - 1000$ GeV, it is induced by $-\lambda_S \simeq 0.05 - 1.26$.[7] For lower masses, $m_{\tilde{\chi}^0_1} \simeq 40$ GeV or $m_{\tilde{\chi}^0_1} \simeq 60$ GeV, where the DM annihilation proceeds via the $Z$ or $h$ pole, and we have $\Delta m \simeq 0.4 - 1.7$ GeV and $|\lambda_S| \simeq 6 \times 10^{-4} - 0.26$ (with $\lambda_S \simeq -0.26$ to 0.02). With the exception of the funnel region, all the bino-like points in the left panel of Figure 7 also have a $\tilde{\chi}^\pm_1$ and $\tilde{\chi}^0_{3,4}$ close in mass to the $\tilde{\chi}^0_1$. This is shown explicitly in the right panel of the same figure. Concretely, we have $m_{\tilde{\chi}^\pm_1} - m_{\tilde{\chi}^0_1} \lesssim 30$ GeV and $m_{\tilde{\chi}^0_{3,4}} - m_{\tilde{\chi}^0_1} \approx 10\text{–}60$ GeV. Often, that is when the LSP has a small wino admixture, the $\tilde{\chi}^\pm_2$ is also close in mass. In most cases $m_{\tilde{\chi}^\pm_1} < m_{\tilde{\chi}^0_3}$ although the opposite case also occurs. All in all this creates peculiar compressed EW-ino spectra; they are similar to the bino-wino DM scenario in the MSSM, but there are more states involved and the possible mass splittings are somewhat larger. In any case, the dominant signatures are 3-body and/or radiative decays of heavier into lighter EW-inos; only the heavier $\tilde{\chi}^\pm_{2,3}$ and $\tilde{\chi}^0_{5,6}$ can decay via an on-shell $W$, $Z$ or $h^0$.

Finally we show in Figure 8 the spin-independent ($\sigma^{\rm SI}$) and spin-dependent ($\sigma^{\rm SD}$) $\tilde{\chi}^0_1$ scattering cross sections on protons, with the $p$-value from XENON1T indicated in colour. While the bulk of the points has cross sections that should be testable in future DM direct

---

[7]Our conventions differ (as usual) from the `SARAH DiracGauginos` implementation: $\lambda_S \equiv -\,$`lam` and $\lambda_T \equiv$ `LT`$/\sqrt{2}$ in `SARAH` convention.

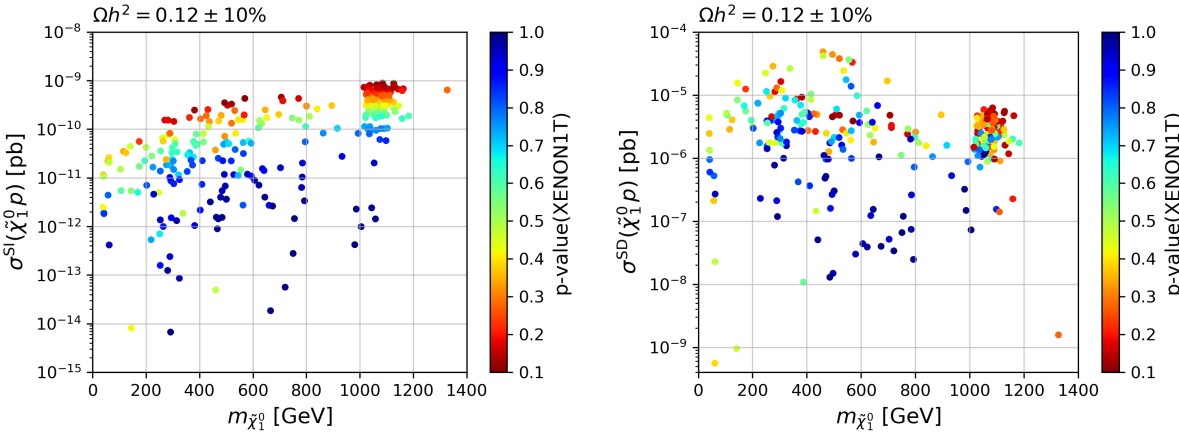

Figure 8: Spin-independent (left) and spin-dependent (right) $\tilde{\chi}_1^0$ scattering cross sections on protons as function of the $\tilde{\chi}_1^0$ mass, for the points with $\Omega h^2 = 0.12 \pm 10\%$. The colour code indicates the $p$-value for XENON1T.

detection experiments, there are also a few points with cross sections below the neutrino floor. We note in passing that the scattering cross section on neutrons (not shown) is not exactly the same in this model but can differ from that on protons by few percent.

## 4.2 LHC constraints

Let us now turn to the question of how the DG EW-ino scenarios from the previous subsection can be constrained at the LHC. Before reinterpreting various ATLAS and CMS SUSY searches, it is important to point out that the cross sections for EW-ino production are larger in the MDGSSM than in the MSSM. For illustration, Figure 9 compares the production cross sections for $pp$ collisions at 13 TeV in the two models. The cross sections are shown as a function of the wino mass parameter, with $m_{D2} = 1.2\, m_{DY}$ ($M_2 = 1.2\, M_1$) for the MDGSSM (MSSM); the other parameters are $\mu \simeq 1400$ GeV, $\tan\beta \simeq 10$, $\lambda_S \simeq -0.29$ and $\sqrt{2}\lambda_T \simeq -1.40$. While LSP-LSP production is almost the same in the two models, chargino-neutralino and chargino-chargino production is about a factor 3–5 larger in the MDGSSM, due to the larger number of degrees of freedom.

### 4.2.1 Constraints from prompt searches

**SModelS**

We start by checking the constraints from searches for promptly decaying new particles with SModelS [102–105]. The working principle of SModelS is to decompose all signatures occurring in a given model or scenario into simplified model topologies, also referred to as simplified model spectra (SMS). Each SMS is defined by the masses of the BSM states, the vertex structure, and the SM and BSM final states. After this decomposition, the signal weights, determined in terms of cross-sections times branching ratios, $\sigma \times \mathrm{BR}$, are matched against a database of LHC results. SModelS reports its results in the form of $r$-values, defined as the ratio of the theory prediction over the observed upper limit, for each experimental constraint

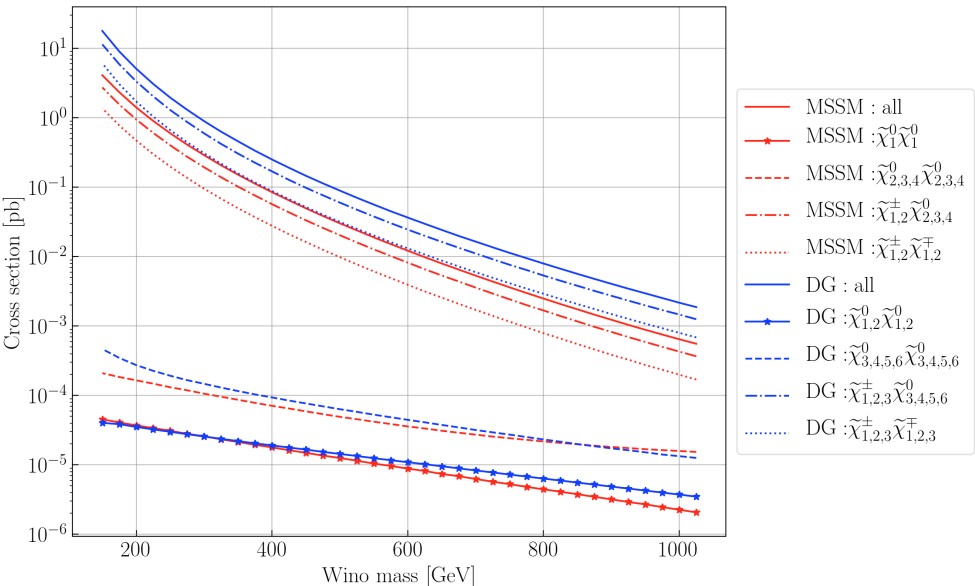

Figure 9:   EW-ino production cross sections at the 13 TeV LHC as a function of the wino mass parameter, in blue for the MDGSSM and in red for the MSSM; the ratio of the bino and wino mass parameters is fixed as $m_{D2} = 1.2\, m_{DY}$ (MDGSSM) and $M_2 = 1.2\, M_1$ (MSSM), while $\mu \simeq 1400$ GeV, $\tan\beta \simeq 10$, $\lambda_S \simeq -0.29$ and $\sqrt{2}\lambda_T \simeq -1.40$.

that is matched in the database. All points for which at least one $r$-value equals or exceeds unity ($r_{\mathrm{max}} \geq 1$) are considered as excluded.

Concretely we are using SModelS v1.2.3 [105]. For our purpose, the most relevant "prompt" search results from Run 2 included in the v1.2.3 database are those from

- the ATLAS EW-ino searches with 139 fb$^{-1}$, constraining $WZ^{(*)} + E_T^{\mathrm{miss}}$ (ATLAS-SUSY-2018-06 [106]), $WH + E_T^{\mathrm{miss}}$ (ATLAS-SUSY-2019-08 [107]) and $WW^{(*)} + E_T^{\mathrm{miss}}$ (ATLAS-SUSY-2018-32 [108]) signatures arising from chargino-neutralino or chargino-chargino production, as well as

- the CMS EW-ino combination for 35.9 fb$^{-1}$, CMS-SUS-17-004 [109], constraining $WZ^{(*)} + E_T^{\mathrm{miss}}$ and $WH + E_T^{\mathrm{miss}}$ signatures from chargino-neutralino production.

One modification we made to the SModelS v1.2.3 database is that we included the combined $WZ^{(*)} + E_T^{\mathrm{miss}}$ constraints from Fig. 8a of [109]; the original v1.2.3 release has only those from Fig. 7a, which are weaker. It is interesting to note that the CMS combination [109] for 35.9 fb$^{-1}$ sometimes still gives stronger limits than the individual ATLAS analyses [106–108] for full Run 2 luminosity.

The SLHA files produced with SPheno in our MCMC scan contain the mass spectrum and decay tables. For evaluating the simplified model constraints with SModelS, also the LHC cross sections at $\sqrt{s} = 8$ and 13 TeV are needed. They are conveniently added to the SLHA files by means of the SModelS–micrOMEGAs interface [90], which moreover automatically produces the correct `particles.py` file to declare the even and odd particle content for SModelS. Once the cross sections are computed, the evaluation of LHC constraints in SModelS takes a few seconds per point, which makes it possible to check the full dataset of 52.5k scan points.

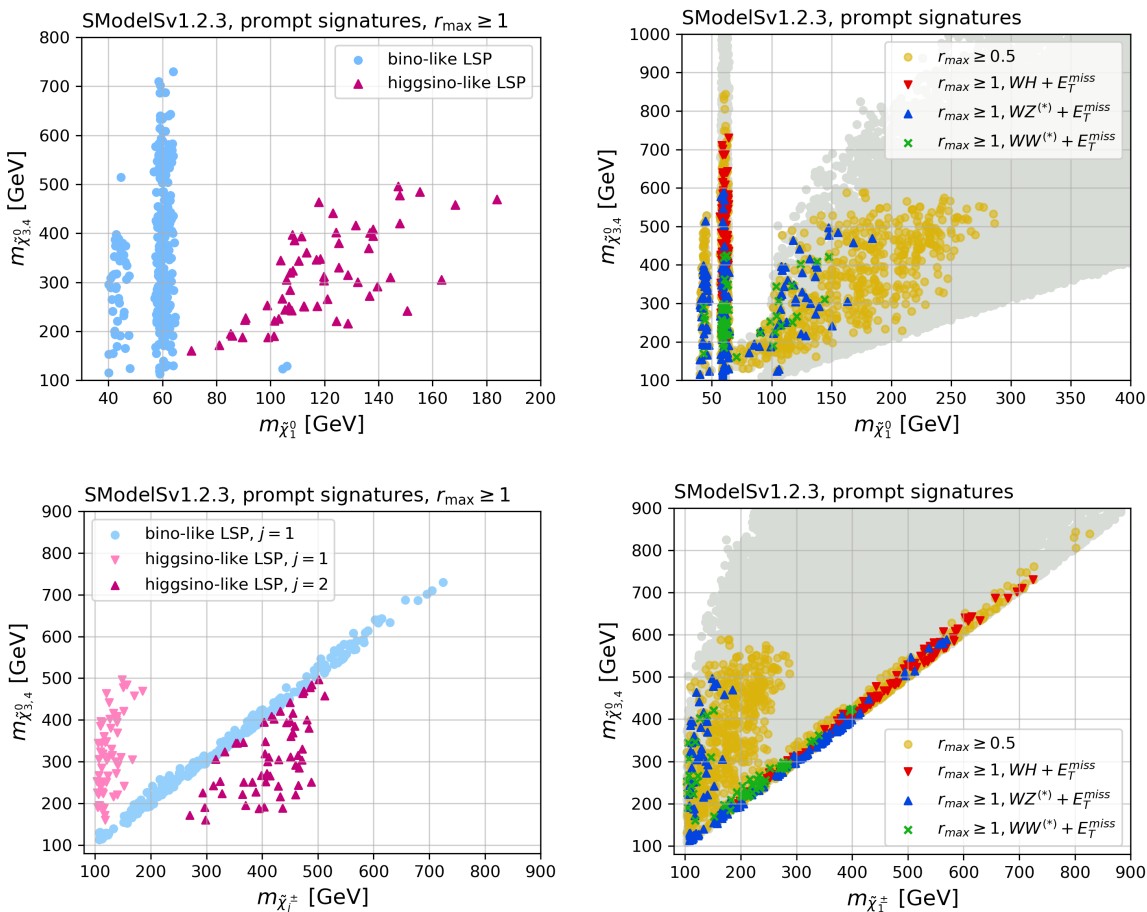

Figure 10: LHC constraints from prompt searches evaluated with SModelS. The left panels show the excluded points, $r_{\max} \geq 1$, in the $m_{\tilde{\chi}_1^0}$ vs. $m_{\tilde{\chi}_{3,4}^0}$ (top) and $m_{\tilde{\chi}_j^\pm}$ vs. $m_{\tilde{\chi}_{3,4}^0}$ (bottom) planes, with bino-like or higgsino-like LSP points distinguished by different colours and symbols as indicated in the plot labels. The right panels show the same mass planes but distinguish the signatures, which are responsible for the exclusion, by different colours/symbols (again, see plot labels); moreover the region with $r_{\max} \geq 0.5$ is shown in yellow, and that covered by all scan points in grey.

The results are shown in Figures 10 and 11. The left panels in Figure 10 show the points excluded by SModelS ($r_{\max} \geq 1$), in the plane of $m_{\tilde{\chi}_1^0}$ vs. $m_{\tilde{\chi}_{3,4}^0}$ (top left) and $m_{\tilde{\chi}_j^\pm}$ vs. $m_{\tilde{\chi}_{3,4}^0}$ (bottom left), the difference between $\tilde{\chi}_{3,4}^0$ not being discernible on the plots. Points with bino-like or higgsino-like LSPs are distinguished by different colours and symbols: light blue dots for bino-like LSP points and magenta/pink triangles for higgsino-like LSP points. There are no excluded points with wino-like LSPs.

As can be seen, apart from two exceptions, all bino LSP points excluded by SModelS lie in the $Z$ or $h$ funnel region and have almost mass-degenerate $\tilde{\chi}_{3,4}^0$ and $\tilde{\chi}_1^\pm$ — actually most of the time they have mass-degenerate $\tilde{\chi}_{3,4}^0$ and $\tilde{\chi}_{1,2}^\pm$ corresponding to a quadruplet of wino states, as winos have much higher production cross sections than higgsinos. The reach is up to about 750 GeV for wino-like $\tilde{\chi}_{3,4}^0$, $\tilde{\chi}_{1,2}^\pm$. When the next-to-lightest states are higgsinos and

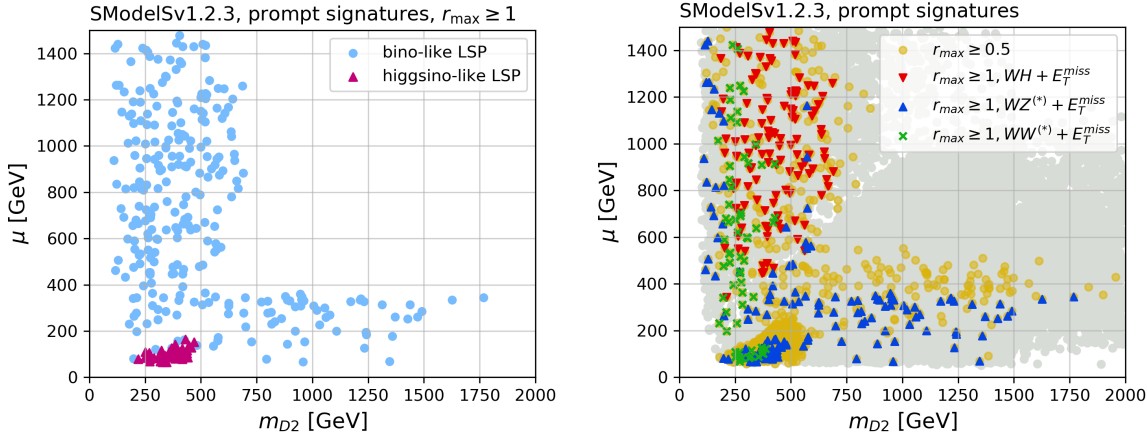

Figure 11:  As Figure 10 but in the $m_{D2}$ vs. $\mu$ plane.

winos are heavy, the exclusion reaches only $m_{\tilde{\chi}^0_{3,4}}$, $m_{\tilde{\chi}^\pm_1} \lesssim 400$ GeV.

The higgsino LSP points excluded by SModelS have $\tilde{\chi}^0_{1,2}$ and $\tilde{\chi}^\pm_1$ masses up to about 200 GeV and always feature light winos ($\tilde{\chi}^0_{3,4}$, $\tilde{\chi}^\pm_{2,3}$) below about 500 GeV. In terms of soft terms, the excluded bino LSP points have $m_{D2} < 750$ GeV or $\mu < 400$ GeV, while the excluded higgsino LSP points have $\mu < 200$ GeV and $m_{D2} < 500$ GeV (see Figure 11).

The right panels of Figures 10 and 11 show the same mass and parameter planes as the left panels but distinguish the signatures, which are responsible for the exclusion, by different colours/symbols. We see that $WH + E_T^{\mathrm{miss}}$ simplified model results exclude only bino-LSP points in the $h$-funnel region, but can reach up to $m_{\tilde{\chi}^0_{3,4}} \lesssim 750$ GeV; all these points have $m_{DY} \approx 60$ GeV, $m_{D2} \lesssim 750$ GeV and $\mu \gtrsim m_{D2}$, cf. Figure 11 (right). The $WZ^{(*)} + E_T^{\mathrm{miss}}$ ($WW^{(*)} + E_T^{\mathrm{miss}}$) simplified model results exclude bino-LSP points in the $Z$- and $h$-funnel regions for winos up to roughly 600 (400) GeV, and higgsino-LSP points with masses up to roughly 200 (150) GeV when the wino-like states are below 500 (400) GeV. Correspondingly, in Figure 11 (right) the green crosses lie in the range $m_{D2} \lesssim 500$ GeV, while blue triangles lie in the region of $m_{D2} \lesssim 600$ GeV or $\mu \lesssim 400$ GeV.

For completeness, the right panels of Figures 10 and 11 also show the region with $r_{\max} \geq 0.5$. This is primarily to indicate how the reach might improve with, e.g., more statistics. It also serves to illustrate the effect of a possible underestimation of the visible signal in the SMS approach, although in the comparison with MadAnalysis 5 below we will see that the limits from simplified models and full recasting actually agree quite well.

We note that we have run SModelS with the default configuration of sigmacut=0.01 fb, minmassgap=5 GeV and maxcond=0.2. Long-lived $\tilde{\chi}^0_2$ are always treated as $E_T^{\mathrm{miss}}$ irrespective of the actual decay length, as the $\tilde{\chi}^0_2 \to \tilde{\chi}^0_1 + X$ decays ($X$ mostly being a photon) are too soft to be picked up/vetoed by the signal selections of the analyses under consideration.[8] The excluded regions depend only slightly on these choices. Overall the constraints are very weak: of the almost 53k scan points, only 340 are excluded by the prompt search results in SModelS; 548 (1126) points have $r_{\max} > 0.8$ (0.5).

---

[8]To this end, we added `if abs(pid) == 1000023:  width = 0.0*GeV` in the `getPromptDecays()` function of `slhaDecomposer.py`; this avoids setting the $\tilde{\chi}^0_2$ decay widths to zero in the input SLHA files.

**MadAnalysis 5**

One disadvantage of the simplified model constraints is that they assume that charginos and neutralinos leading to $WZ^{(*)} + E_T^{\mathrm{miss}}$ or $WH + E_T^{\mathrm{miss}}$ signatures are mass degenerate. SModelS allows a small deviation from this assumption, but $\tilde{\chi}_i^{\pm} \tilde{\chi}_j^0$ production with sizeable differences between $m_{\tilde{\chi}_i^{\pm}}$ and $m_{\tilde{\chi}_j^0}$ will not be constrained. Moreover, the simplified model results from [106–109] are cross section upper limits only, which means that different contributions to the same signal region cannot be combined (to that end efficiency maps would be necessary [103]). It is therefore interesting to check whether full recasting based on Monte Carlo event simulation can extend the limits derived with SModelS.

   Here we use the recast codes [110–112] for Run 2 EW-ino searches available in MadAnalysis 5 [113–116].[9] These are

- two CMS searches in leptons $+E_T^{\mathrm{miss}}$ final states for 35.9 fb$^{-1}$ of Run 2 data, namely the multi-lepton analysis CMS-SUS-16-039 [117], for which the combination of signal regions via the simplified likelihood approach has recently been implemented in MadAnalysis 5 (see contribution no. 15 in [118]), and the soft lepton analysis CMS-SUS-16-048 [119], which targets compressed EW-inos; as well as

- the ATLAS search in the $1l + H(\to b\bar{b}) + E_T^{\mathrm{miss}}$ final state based on 139 fb$^{-1}$ of data, ATLAS-SUSY-2019-08 [107], which targets the $WH + E_T^{\mathrm{miss}}$ channel and which we newly implemented for this study (details are given in appendix A.4).

   For these analyses we again treat the two lightest neutralino states as LSPs, assuming the transition $\tilde{\chi}_2^0 \to \tilde{\chi}_1^0$ is too soft as to be visible in the detector. For the CMS 35.9 fb$^{-1}$ analyses, we simulate all possible combinations of $\tilde{\chi}_{1,2}^0$ with the heavy neutralinos, charginos, and pair production of charginos; while to recast the analysis of [107] we must simulate $pp \to \tilde{\chi}_i^{\pm} \tilde{\chi}_{j>2}^0 + n$jets, where $n$ is between zero and two. The hard process is simulated in MadGraph5_aMC@NLO [120] v2.6 and passed to Pythia 8.2 [121] for showering. MadAnalysis 5 handles the detector simulation with Delphes 3 [122] with different cards for each analysis, and then computes exclusion confidence levels $(1 - \mathrm{CL}_s)$, including the combination of signal regions for the multi-lepton analysis. For the two 35.9 fb$^{-1}$ analyses we simulate 50k events, and the whole simulation takes more than an hour per point on an 8-core desktop PC. For the ATLAS 139 fb$^{-1}$ analysis, we simulate 100k events (because of the loss of efficiency in merging jets, and targeting only $b$-jets from the Higgs and in particular the leptonic decay channel of the $W$) and each point requires 3 hours.

   The reach of collider searches depends greatly on the wino fraction of the EW-inos. Winos have a much higher production cross section than higgsinos or binos, and thus we can divide the scan points into those where $m_{D2}$ is "light" and "heavy." The results are shown in Figure 12. They show the distribution of points in our scan in the $m_{\tilde{\chi}_1^0} - m_{\tilde{\chi}_3^0}$ plane. In our model, there is always a pseudo-Dirac LSP, so the lightest neutralinos are nearly degenerate; for a higgsino- or wino-like LSP the lightest chargino is nearly degenerate with the LSP. However, $m_{\tilde{\chi}_3^0}$ gives the location of the next lightest states, irrespective of the LSP type. In this plane we show the points that we tested using MadAnalysis 5, and delineate the region encompassing all excluded points.

   For "light" $m_{D2} < 900$ GeV, nearly all tested points in the Higgs funnel are excluded by [107] up to $m_{\tilde{\chi}_3} = 800$ GeV; the $Z$-funnel is excluded for $m_{\tilde{\chi}_3} \lesssim 300$ GeV. Otherwise we can

---

[9]See http://madanalysis.irmp.ucl.ac.be/wiki/PublicAnalysisDatabase.

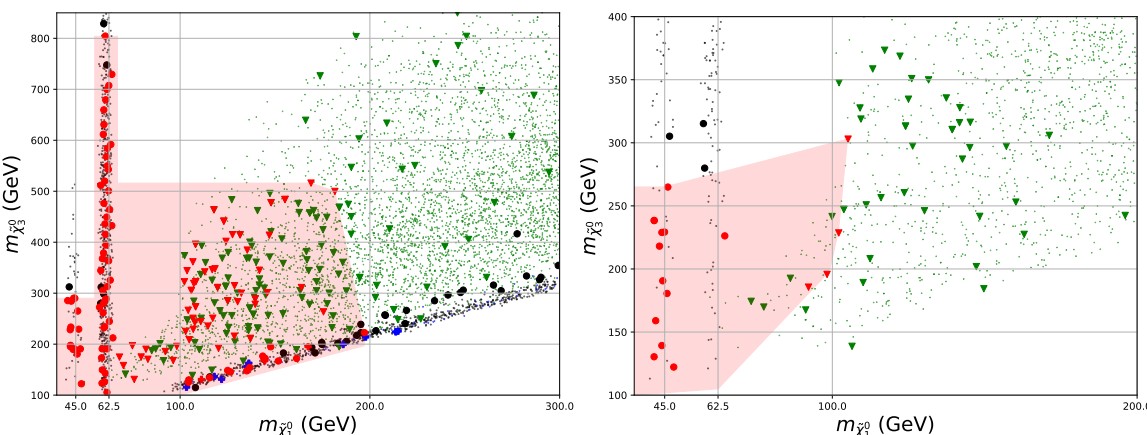

Figure 12: DM-compatible points found in our scan ($\Omega h^2 \leq 0.132$) in the plane of lightest neutralino vs. third lightest neutralino mass. The left plot shows points for which $m_{D2} < 900$ GeV, the right plot has $m_{D2} > 700$ GeV. Higgsino-like LSP points are shown in green, winos in blue and binos in black. The red transparent region surrounds all points that were found to be excluded using MadAnalysis 5; the location of the recast points are shown as large circles (binos), crosses (winos) and triangles (higgsinos). Excluded points are coloured red.

find excluded points in the region $m_{\tilde\chi^0_1} \lesssim 200$ GeV, $m_{\tilde\chi^0_3} \lesssim 520$ GeV. While for small $m_{\tilde\chi^0_3} - m_{\tilde\chi^0_1}$ the ATLAS-SUSY-2019-08 search [107] is not effective, at large values of $m_{\tilde\chi^0_3}$ some points are excluded by this analysis, and others still by CMS-SUS-16-039 [117] and/or CMS-SUS-16-048 [119]. We note here that the availability of the covariance matrix for signal regions A of [117] is quite crucial for achieving a good sensitivity. It would be highly beneficial to have more such (full or simplified) likelihood data that allows for the combination of signal regions!

For "heavy" $m_{D2} > 700$ GeV,[10] we barely constrain the model at all: clearly $Z$-funnel points are excluded up to about $m_{\tilde\chi^0_3} = 260$ GeV; but we only find excluded points for $m_{\tilde\chi^0_1} \lesssim 100$ GeV, $m_{\tilde\chi^0_3} \lesssim 300$ GeV. Hence one of the main conclusions of this work is that higgsino/bino mixtures in this model, where $m_{D2} > 700$ GeV, are essentially unconstrained for $m_{\tilde\chi^0_1} \gtrsim 120$ GeV.

In general, as in [69], one may expect a full recast in MadAnalysis 5 to be much more powerful than a simplified models approach. However, comparing the results from MadAnalysis 5 to those from SModelS, a surprisingly good agreement is found between the $r$-values from like searches (such as the $WH + E_T^{\rm miss}$ channel in the same analysis).[11] Indeed, from comparing Figures 12 with the upper two panels in Figure 10, we see that the excluded region is very similar, with perhaps a small advantage to the full MadAnalysis 5 recasting at the top of the Higgs funnel and at larger values of $m_{\tilde\chi^0_3}$ for higgsino LSPs, while SModelS (partly thanks to more 139 fb$^{-1}$ analyses) is more powerful in the $Z$-funnel region. A detailed comparison leads to the following observations:

- The $WZ + E_T^{\rm miss}$ upper limits in SModelS can be more powerful than the recasting of the individual analyses implemented in MadAnalysis 5. As an example, consider the two

---

[10] The regions are only not disjoint so that we can include the entire constrained reach of the Higgs funnel in the "light" plot; away from the Higgs funnel there would be no difference in the "light" $m_{D2}$ plot if we took $m_{D2} < 700$ GeV.

[11] We shall see this explicitly for some benchmark scenarios in section 5.

neighbouring points with $(m_{DY}, m_{D2}, \mu, \tan\beta, -\lambda_S, \sqrt{2}\lambda_T) = (742.6, 435.7, 164.1, 5.83, 0.751, 0.491)$ and $(746.6, 459.9, 154.2, 12.77, 0.846, 0.466)$, with mass parameters in GeV units. They respectively have $(m_{\tilde\chi_1^0}, m_{\tilde\chi_3^0}, m_{\tilde\chi_5^0}) = (189, 474, 753)$ GeV and $(182, 500, 761)$ GeV, i.e. well spread spectra with higgsino LSPs. For the first point SModelS gives $r_{\max} = 0.99$ and for the second $r_{\max} = 0.84$ from the CMS EW-ino combination [109]. The $1 - \mathrm{CL}_s$ values from MadAnalysis 5 are 0.79 and 0.84, respectively, from the combination of signal regions A of the CMS multi-lepton search [117]; in terms of the ratio $r_{\mathrm{MA5}}$ of predicted over excluded (visible) cross sections, this corresponds to $r_{\mathrm{MA5}} = 0.67$ and 0.71, so somewhat lower than the values from SModelS.

- The $WH + E_T^{\mathrm{miss}}$ signal for the two example points above splits up into several components (corresponding to different mass vectors) in SModelS, which each give $r$-values of roughly 0.3 but cannot be combined. The recast of ATLAS-SUSY-2019-08 [107] with MadAnalysis 5, on the other hand, takes the complete signal into account and gives $1 - \mathrm{CL}_s = 0.77$ for the first and 0.96 for the second point.

- The points excluded with MadAnalysis 5 but not with SModelS typically contain complex spectra with all EW-inos below about 800 GeV, which all contribute to the signal.

- Most tested points away from the Higgs funnel region, which are excluded with MadAnalysis 5 but not with SModelS, have $r_{\max} > 0.8$.

- There also exist points which are excluded by SModelS but not by the recasting with MadAnalysis 5. In these cases the exclusion typically comes from the CMS EW-ino combination [109]; detailed likelihood information would be needed to emulate this combination in recasting codes.

It would be interesting to revisit these conclusions once more EW-ino analyses are implemented in full recasting tools, but it is clear that, since adding more luminosity does not dramatically alter the constraints, the SModelS approach can be used as a reliable (and much faster) way of constraining the EW-ino sector; and that the constraints on EW-inos in Dirac gaugino models are still rather weak, particularly for higgsino LSPs where the wino is heavy.

### 4.2.2    Constraints from searches for long-lived particles

As mentioned in section 4.1, a relevant fraction (about 20%) of the points in our dataset contain LLPs. Long-lived charginos, which occur in about 14% of all points, can be constrained by Heavy Stable Charged Particles (HSCP) and Disappearing Tracks (DT) searches. Displaced vertex (DV) searches could potentially be sensitive to long-lived neutralinos; in our case however, the decay products of long-lived neutralinos are typically soft photons, and there is no ATLAS or CMS analysis which would be sensitive to these.

We therefore concentrate on constraints from HSCP and DT searches. They can conveniently be treated in the context of simplified models. For HSCP constraints we again use SModelS, which has upper limit and efficiency maps from the full 8 TeV [123] and early 13 TeV (13 fb$^{-1}$) [124] CMS analyses implemented. (The treatment of LLPs in SModelS is described in detail in Refs. [104,125].) A new 13 TeV analysis for 36 fb$^{-1}$ is available from ATLAS [126], but not yet included in SModelS; we will come back to this below.

For the DT case, the ATLAS [127] and CMS [128] analyses for 36 fb$^{-1}$ provide 95% CL upper limits on $\sigma \times \mathrm{BR}$ in terms of chargino mass and lifetime on HEPData [129, 130].

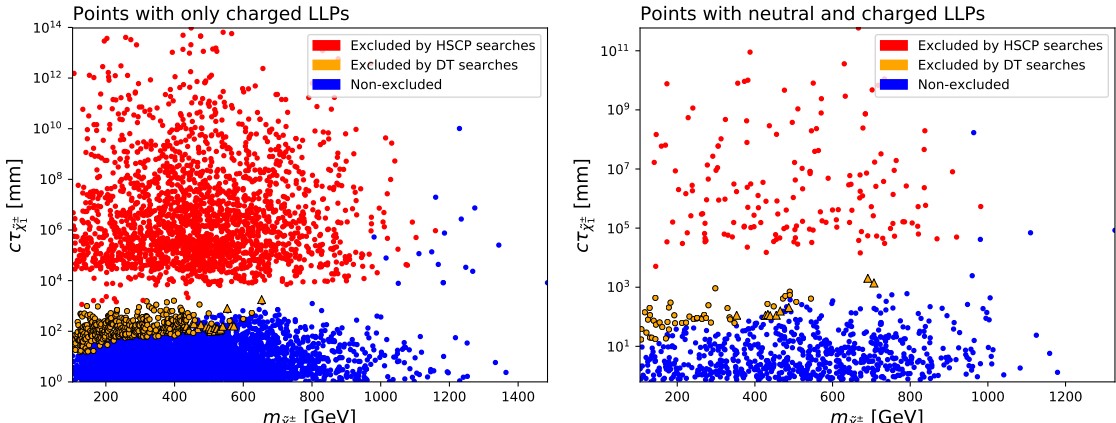

Figure 13: Exclusion plots for points with only charged LLPs (left) and points with neutral and charged LLPs (right), obtained in the simplified model approach. Red points are excluded by the HSCP searches implemented in SModelS, orange points are excluded by DT searches; the latter are plotted as circles if excluded at 36 fb$^{-1}$ and as triangles if excluded at 140 fb$^{-1}$. Non-excluded points are shown in blue.

Here, $\sigma \times$ BR stands for the cross section of direct production of charginos, which includes $\tilde{\chi}_1^{\pm}\tilde{\chi}_1^{\mp}$ and $\tilde{\chi}_1^{\pm}\tilde{\chi}_1^0$ production, times BR($\tilde{\chi}_1^{\pm} \to \tilde{\chi}_1^0\pi^{\pm}$), for each produced chargino. Using the `interpolate.griddata` function from `scipy`, we estimated the corresponding 95% CL upper limits for our scan points within the reach of each analysis[12] from a linear interpolation of the HEPData tables. This was then used to compute $r$-values as the ratio of the predicted signal over the observed upper limit, similar to what is done in SModelS. The points with only charged ($\tilde{\chi}_1^{\pm}$) LLPs and those with both charged and neutral ($\tilde{\chi}_2^0$) LLPs are treated on equal footing. However, for the points which have both a neutral and a charged LLP, if $m_{\tilde{\chi}_1^{\pm}} > m_{\tilde{\chi}_2^0}$, the $\tilde{\chi}_1^{\pm}\tilde{\chi}_2^0$ direct production cross section and the branching fraction of $\tilde{\chi}_1^{\pm} \to \tilde{\chi}_2^0\pi^{\pm}$ were also included.

There is also a new CMS DT analysis [131], which presents full Run 2 results for 140 fb$^{-1}$. At the time of our study, this analysis did not yet provide any auxiliary (numerical) material for reinterpretation. We therefore digitised the limits curves from Figures 1a–1d of that paper, and used them to construct linearly interpolated limit maps which are employed in the same way as described in the previous paragraph. Since the interpolation is based on only four values of chargino lifetimes, $\tau_{\tilde{\chi}_1^{\pm}} = 0.33$, 3.34, 33.4 and 333 ns, this is however less precise than the interpolated limits for 36 fb$^{-1}$.

The results are shown in Figure 13 in the plane of chargino mass vs. mean decay length; on the left for points with long-lived charginos, on the right for point with long-lived charginos and neutralinos. Red points are excluded by the HSCP searches implemented in SModelS: orange points are excluded by DT searches. The HSCP limits from [123, 124] eliminate basically all long-lived chargino scenarios with $c\tau_{\tilde{\chi}^{\pm}} \gtrsim 1$ m up to about 1 TeV chargino mass. The exclusion by the DT searches [127, 128] covers 10 mm $\lesssim c\tau_{\tilde{\chi}_1^{\pm}} \lesssim 1$ m and $m_{\tilde{\chi}_1^{\pm}}$ up to about 600 GeV; this is only slightly extended to higher masses by our reinterpretation of the limits of [131].

---

[12]This is $95 < m_{\tilde{\chi}_1^{\pm}} < 600$ GeV and $0.05 < \tau_{\tilde{\chi}_1^{\pm}} < 4$ ns ($15 < c\tau_{\tilde{\chi}_1^{\pm}} < 1200$ mm) for the ATLAS analysis [127], and $100 < m_{\tilde{\chi}_1^{\pm}} < 900$ GeV and $0.067 < \tau_{\tilde{\chi}_1^{\mp}} < 333.56$ ns ($20 < c\tau_{\tilde{\chi}_1^{\pm}} < 100068$ mm) for the CMS analysis [128].

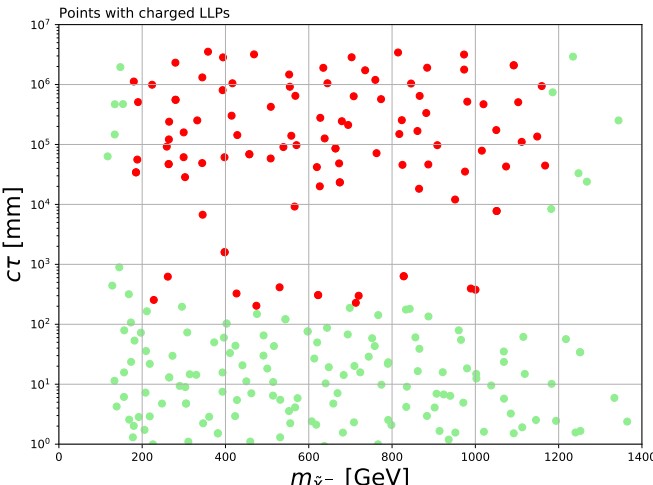

Figure 14: Exclusion for charged LLPs using A. Lessa's recast code for the ATLAS HSCP search [126] from `https://github.com/llprecasting/recastingCodes`; red points are excluded, green points are not excluded by this analysis.

The white band in-between $c\tau \approx 10^3$–$10^4$ mm corresponds to $m_{\tilde\chi_1^\pm} - m_{\tilde\chi_1^0} \approx m_{\pi^\pm}$: the chargino lifetime changes significantly when decays into pions become kinematically forbidden.

To verify the HSCP results from SModelS and extend them to 36 fb$^{-1}$, we adapted the code for recasting the ATLAS analysis [126] written by A. Lessa and hosted at `https://github.com/llprecasting/recastingCodes`. This requires simulating hard processes of single/double chargino LLP production with *two* additional hard jets, which was performed at leading order with MadGraph5_aMC@NLO. The above code then calls Pythia 8.2 to shower and decay the events, and process the cuts. It uses experiment-provided efficiency tables for truth-level events rather than detector simulation, and therefore does not simulate the presence of a magnetic field. However, the code was validated by the original author for the MSSM chargino case and found to give excellent agreement.

We wrote a parallelised version of the recast code to speed up the workflow (which is available upon request); the bottleneck in this case is actually the simulation of the hard process (unlike for the prompt recasting case in the previous section), and our sample was simulated on one desktop. We show the result in Figure 14. For decay lengths $c\tau_{\tilde\chi_1^\pm} > 1$ m, the exclusion is very similar to that from SModelS, only slightly extending it in the $m_{\tilde\chi_1^\pm} \approx 1$–1.2 TeV range. For decay lengths of about 0.2–1 m, the recasting with event simulation allows the exclusion of points in the 0.2–1 TeV mass range; this region is not covered by SModelS. As with the SModelS results, we see that LLP searches are extremely powerful, and where a parameter point contains an LLP with a mass and lifetime in the correct range for a search, there is no possibility to evade exclusion.

## 4.3   Future experiments: MATHUSLA

We also investigated the possibility of seeing events in the MATHUSLA detector [132], which would be built $\mathcal{O}(100)$m from the collision point at the LHC, and so would be able to detect neutral particles that decay after such a long distance. Prima facie this would seem ideal to search for the decays of long-lived neutralino NLSPs; pseudo-Dirac states should be excellent

candidates for this (indeed, the possibility of looking for similar particles if they were of $\mathcal{O}(\text{GeV})$ in mass at the SHiP detector was investigated in [133]). However, in our case the only states that have sufficient lifetime to reach the detector have mass splittings of $\mathcal{O}(10)$ MeV (or less), and decays $\tilde{\chi}_2^0 \to \tilde{\chi}_1^0 + \gamma$ vastly dominate, with a tiny fraction of decays to electrons.

In the detectors in the roof of MATHUSLA the photons must have more than 200 MeV (or 1 GeV for electrons) to be registered. Moreover, it is anticipated to reconstruct the decay vertex in the decay region, requiring more than one track; in our case only one track would appear, and much too soft to trigger a response. Hence, unless new search strategies are employed, our long-lived $\tilde{\chi}_2^0$ will escape detection.

## 5 Benchmark points

In this section we present a few sample points which may serve as benchmarks for further studies, designing dedicated experimental analyses and/or investigating the potential of future experiments. Parameters, masses, and other relevant quantities are listed in Tables 2 and 3.

**Point 1 (`SPhenoDiracGauginos_667`)** lies in the $h$-funnel region. It features almost pure bino $\tilde{\chi}_{1,2}^0$ with masses of 62–63 GeV, higgsino-like $\tilde{\chi}_1^\pm$ and $\tilde{\chi}_{3,4}^0$ with masses around 560–580 GeV, and heavy wino-like $\tilde{\chi}_{5,6}^0$ and $\tilde{\chi}_{2,3}^\pm$ around 1.2 TeV. A relic abundance in accordance with the cosmologically observed value is achieved through $\tilde{\chi}_1^0 \tilde{\chi}_2^0$ co-annihilation into $b\bar{b}$ (63%), $gg$ (17%) and $\tau^+ \tau^-$ (13%) via $s$-channel $h$ exchange.[13] Kinematically just allowed, invisible decays of the Higgs boson have a tiny branching ratio, $\text{BR}(h \to \tilde{\chi}_1^0 \tilde{\chi}_2^0) = 5.2 \times 10^{-4}$, and thus do not affect current Higgs measurements or coupling fits. The main decay modes of the EW-inos are:

| mass | decays |
|---|---|
| 1254 GeV | $\tilde{\chi}_3^\pm \to \tilde{\chi}_1^\pm Z$ (57%), $\tilde{\chi}_1^\pm h$ (42%) |
| 1235 GeV | $\tilde{\chi}_6^0 \to \tilde{\chi}_3^0 Z$ (32%), $\tilde{\chi}_4^0 h$ (29%), $\tilde{\chi}_1^\pm W^\pm$ (36%) |
| 1233 GeV | $\tilde{\chi}_5^0 \to \tilde{\chi}_4^0 Z$ (33%), $\tilde{\chi}_3^0 h$ (30%), $\tilde{\chi}_1^\pm W^\pm$ (36%) |
| 1212 GeV | $\tilde{\chi}_2^\pm \to \tilde{\chi}_3^0 W^\pm$ (49%), $\tilde{\chi}_4^0 W^\pm$ (49%) |
| 584 GeV | $\tilde{\chi}_4^0 \to \tilde{\chi}_1^0 h$ (33%), $\tilde{\chi}_2^0 h$ (25%), $\tilde{\chi}_2^0 Z$ (21%), $\tilde{\chi}_1^0 Z$ (20%) |
| 582 GeV | $\tilde{\chi}_3^0 \to \tilde{\chi}_1^0 Z$ (30%), $\tilde{\chi}_2^0 Z$ (26%), $\tilde{\chi}_2^0 h$ (24%), $\tilde{\chi}_1^0 h$ (20%) |
| 564 GeV | $\tilde{\chi}_1^\pm \to \tilde{\chi}_1^0 W^\pm$ (51%), $\tilde{\chi}_2^0 W^\pm$ (48%) |
| 63 GeV | $\tilde{\chi}_2^0 \to \tilde{\chi}_1^0 \gamma$ (86%); $\Gamma_{\text{tot}} = 6.6 \times 10^{-17}$ GeV ($c\tau \approx 3$ m) |
| 62 GeV | $\tilde{\chi}_1^0$, stable |

Regarding LHC signals, $pp \to \tilde{\chi}_1^\pm \tilde{\chi}_{3,4}^0$ production has a cross section of about 9 fb at $\sqrt{s} = 13$ TeV and leads to almost equal rates of $WZ + E_T^{\text{miss}}$ and $WH + E_T^{\text{miss}}$ ($H \equiv h$) signatures, accompanied by soft displaced photons in 3/4 of the cases. With $\tilde{\chi}_{3,4}^0$ masses only 1.7 GeV apart,

---

[13]This is one example where the precise calculation of the NLSP decays influences the value of the relic density. Without the $\tilde{\chi}_2^0 \to \tilde{\chi}_1^0 \gamma$ loop calculation, $\Gamma_{\text{tot}}(\tilde{\chi}_2^0) = 9 \times 10^{-18}$ GeV and $\Omega h^2 = 0.111$. Including the loop decay, we get $\Gamma_{\text{tot}}(\tilde{\chi}_2^0) = 6.6 \times 10^{-17}$ GeV and $\Omega h^2 = 0.127$. Note also that one has to set `useSLHAwidth=1` in micrOMEGAs to reproduce these values with SLHA file input.

| Point | 1 | 2 | 3 | 4 | 5 |
|---|---|---|---|---|---|
| $m_{DY}$ | 62.58 | 184.24 | 553.94 | 555.47 | 382.20 |
| $m_{D2}$ | 1170.19 | 221.81 | 553.59 | 602.61 | 594.06 |
| $\mu$ | 605.67 | 1454.11 | 1481.55 | 1115.58 | 480.55 |
| $\tan\beta$ | 15.63 | 10.44 | 7.92 | 12.28 | 28.05 |
| $-\lambda_S$ | 0.016 | 1.13 | 0.97 | 0.60 | 0.27 |
| $\sqrt{2}\lambda_T$ | $-1.26$ | $-0.86$ | 0.07 | $-1.2$ | $-0.93$ |
| $m_{\tilde\chi_1^0}$ | 62.34 | 195.23 | 561.69 | 563.82 | 387.74 |
| $m_{\tilde\chi_2^0}$ | 63.45 | 211.70 | 576.12 | 568.31 | 387.92 |
| $m_{\tilde\chi_3^0}$ | 581.86 | 222.47 | 589.85 | 600.39 | 432.96 |
| $m_{\tilde\chi_4^0}$ | 583.62 | 224.13 | 592.91 | 606.63 | 433.87 |
| $m_{\tilde\chi_5^0}$ | 1233.07 | 1523.80 | 1532.71 | 1162.02 | 669.12 |
| $m_{\tilde\chi_6^0}$ | 1234.85 | 1528.71 | 1536.34 | 1166.42 | 669.53 |
| $m_{\tilde\chi_1^\pm}$ | 563.75 | 215.00 | 588.28 | 580.86 | 398.60 |
| $m_{\tilde\chi_2^\pm}$ | 1212.35 | 229.86 | 592.69 | 626.84 | 619.96 |
| $m_{\tilde\chi_3^\pm}$ | 1254.34 | 1521.61 | 1527.55 | 1184.63 | 703.47 |
| $f_{\tilde b}$ | 0.997 | 0.95 | 0.97 | 0.96 | 0.997 |
| $f_{\tilde w}$ | $O(10^{-5})$ | 0.04 | 0.02 | 0.03 | $O(10^{-5})$ |
| $f_{\tilde h}$ | $O(10^{-3})$ | 0.01 | 0.01 | 0.01 | $O(10^{-3})$ |
| $\Omega h^2$ | 0.127 | 0.116 | 0.127 | 0.127 | 0.113 |
| $\sigma^{\mathrm{SI}}(\tilde\chi_1^0 p)$ | $9.4\times10^{-13}$ | $2.2\times10^{-11}$ | $1.6\times10^{-10}$ | $1.2\times10^{-10}$ | $1.8\times10^{-10}$ |
| $\sigma^{\mathrm{SD}}(\tilde\chi_1^0 p)$ | $2.7\times10^{-7}$ | $4\times10^{-6}$ | $1.9\times10^{-6}$ | $2.7\times10^{-6}$ | $1.1\times10^{-8}$ |
| $p_{X1T}$ | 0.93 | 0.62 | 0.42 | 0.50 | 0.29 |
| $r_{\max}$ | 0.39 | $-$ | $-$ | $-$ | $-$ |
| $1-\mathrm{CL}_s$ | 0.65 | 0.51 | 0.02 | 0.03 | 0.07 |
| $\sigma_{\mathrm{LHC13}}$ | 14.9 | 2581 | 41.2 | 35.9 | 87.8 |
| $\sigma_{\mathrm{LHC14}}$ | 18.0 | 2910 | 49.6 | 43.8 | 103.1 |

Table 2: Overview of benchmark points 1–5. Masses and mass parameters are in GeV, $\tilde\chi_1^0 p$ scattering cross sections in pb, and LHC cross sections in fb units. $f_{\tilde b}$, $f_{\tilde w}$ and $f_{\tilde h}$ are the bino, wino and higgsino fractions of the $\tilde\chi_1^0$, respectively. $r_{\max}$ is the highest $r$-value from SModelS (when relevant), while $1-\mathrm{CL}_s$ is the exclusion CL from MadAnalysis 5. $\sigma_{\mathrm{LHC13}}$ and $\sigma_{\mathrm{LHC14}}$ are the total EW-ino production cross sections (sum over all channels) at 13 and 14 TeV computed with MadGraph5_aMC@NLO; the statistical uncertainties on these cross sections are 3% for Point 2, and about 5–7% otherwise.

| Point | 6 | 7 | 8 | 9 | 10 |
|---|---|---|---|---|---|
| $m_{DY}$ | 1452.39 | 1919.27 | 1304.08 | 1365.50 | 809.67 |
| $m_{D2}$ | 1459.01 | 1229.16 | 1269.15 | 848.28 | 446.83 |
| $\mu$ | 1033.56 | 1105.53 | 1957.19 | 572.96 | 224.68 |
| $\tan\beta$ | 7.67 | 17.17 | 33.24 | 9.57 | 6.05 |
| $-\lambda_S$ | 0.81 | 1.10 | 1.39 | 0.90 | 0.81 |
| $\sqrt{2}\lambda_T$ | 0.42 | 0.29 | 0.05 | 0.31 | 0.37 |
| $m_{\tilde\chi_1^0}$ | 1075.01 | 1158.96 | 1327.19 | 605.27 | 246.93 |
| $m_{\tilde\chi_2^0}$ | 1079.15 | 1159.09 | 1327.31 | 605.71 | 247.19 |
| $m_{\tilde\chi_3^0}$ | 1470.39 | 1295.59 | 1346.21 | 900.98 | 484.79 |
| $m_{\tilde\chi_4^0}$ | 1473.61 | 1296.08 | 1356.92 | 901.04 | 485.79 |
| $m_{\tilde\chi_5^0}$ | 1527.23 | 1951.32 | 2076.15 | 1380.78 | 821.83 |
| $m_{\tilde\chi_6^0}$ | 1528.27 | 1957.08 | 2078.22 | 1383.37 | 821.86 |
| $m_{\tilde\chi_1^\pm}$ | 1081.00 | 1159.38 | 1327.28 | 605.50 | 247.28 |
| $m_{\tilde\chi_2^\pm}$ | 1526.26 | 1291.71 | 1331.70 | 898.31 | 480.35 |
| $m_{\tilde\chi_3^\pm}$ | 1528.71 | 1299.64 | 2059.14 | 903.81 | 490.70 |
| $f_{\tilde b}$ | 0.02 | 0.01 | 0.05 | 0.01 | 0.02 |
| $f_{\tilde w}$ | $O(10^{-4})$ | 0.03 | 0.94 | $O(10^{-3})$ | 0.01 |
| $f_{\tilde h}$ | 0.98 | 0.96 | 0.01 | 0.99 | 0.97 |
| $\Omega h^2$ | 0.112 | 0.124 | 0.11 | 0.04 | 0.006 |
| $\sigma^{\rm SI}(\tilde\chi_1^0 p)$ | $4.1\times10^{-10}$ | $6.2\times10^{-10}$ | $6.4\times10^{-10}$ | $5.6\times10^{-11}$ | $1.2\times10^{-9}$ |
| $\sigma^{\rm SD}(\tilde\chi_1^0 p)$ | $4.2\times10^{-6}$ | $2.3\times10^{-7}$ | $1.6\times10^{-9}$ | $1.3\times10^{-6}$ | $2.1\times10^{-5}$ |
| $p_{X1T}$ | 0.35 | 0.20 | 0.28 | 0.92 | 0.46 |
| $r_{\max}$ | – | – | 0.28 | – | 0.39 |
| $1-{\rm CL}_s$ | – | – | – | – | 0.73 |
| $\sigma_{\rm LHC13}$ | 0.48 | 0.65 | 0.32 | 13.2 | 490.5 |
| $\sigma_{\rm LHC14}$ | 0.64 | 0.90 | 0.45 | 16.3 | 557.3 |

Table 3: Overview of benchmark points 6–10. Notation and units as in Table 2. The statistical uncertainties on the LHC cross sections are about 10% for Points 6–8, 6–7% for Point 9 and 3–4% for Point 10.

SModelS adds up signal contributions from $\tilde{\chi}_1^\pm \tilde{\chi}_3^0$ and $\tilde{\chi}_1^\pm \tilde{\chi}_4^0$ production. This gives $r$-values of about 0.4 for the $WH + E_T^{\mathrm{miss}}$ topology (ATLAS-SUSY-2019-08 [107]) and about 0.3 for the $WZ + E_T^{\mathrm{miss}}$ topology (CMS-SUS-17-004 [109] and ATLAS-SUSY-2017-03 [134])[14] in good agreement with the exclusion confidence level (CL), $1 - \mathrm{CL}_s = 0.645$, obtained with MadAnalysis 5 from recasting ATLAS-SUSY-2019-08 [107], and $1 - \mathrm{CL}_s = 0.26$ from the combination of signal regions A from CMS-SUS-16-039 [117].

**Point 2 (SPhenoDiracGauginos_50075)** has a $\tilde{\chi}_1^0$ mass of 195 GeV and a large $\tilde{\chi}_1^0$–$\tilde{\chi}_2^0$ mass difference of 16 GeV due to $\lambda_S = -1.13$. The LSP is 95% bino and 4% wino. The next-lightest states are the wino-like $\tilde{\chi}_{1,2}^\pm$ and $\tilde{\chi}_{3,4}^0$ with masses of 215–230 GeV ($m_{\tilde{\chi}_1^\pm} < m_{\tilde{\chi}_{3,4}^0} < m_{\tilde{\chi}_2^\pm}$). The higgsino-like $\tilde{\chi}_3^\pm$ and $\tilde{\chi}_{5,6}^0$ are heavy with masses around 1.5 TeV. A relic density of the right order, $\Omega h^2 = 0.116$, is achieved primarily through co-annihilations, in particular $\tilde{\chi}_1^0 \tilde{\chi}_1^\pm$ (29%) and $\tilde{\chi}_1^+ \tilde{\chi}_1^-$ (20%) co-annihilation into a large variety of final states; the main LSP pair-annihilation channel is $\tilde{\chi}_1^0 \tilde{\chi}_1^0 \rightarrow W^+ W^-$ and contributes 15%. The main decay modes relevant for collider signatures are:

| mass | decays |
| --- | --- |
| 230 GeV | $\tilde{\chi}_2^\pm \rightarrow \tilde{\chi}_1^0 W^*$ (82%), $\tilde{\chi}_1^\pm \gamma$ (11%) |
| 220 GeV | $\tilde{\chi}_{3,4}^0 \rightarrow \tilde{\chi}_1^\pm W^*$ (98–99%), $\tilde{\chi}_1^0 \gamma$ (2–1%) |
| 215 GeV | $\tilde{\chi}_1^\pm \rightarrow \tilde{\chi}_1^0 W^*$ (100%) |
| 212 GeV | $\tilde{\chi}_2^0 \rightarrow \tilde{\chi}_1^0 \gamma$ (87%), $\tilde{\chi}_1^0 Z^*$ (13%); $\Gamma_{\mathrm{tot}} = 8.2 \times 10^{-10}$ GeV (prompt) |
| 195 GeV | $\tilde{\chi}_1^0$, stable |

Despite the large cross section for $\tilde{\chi}_{1,2}^\pm \tilde{\chi}_{3,4}^0$ ($\tilde{\chi}_{1,2}^+ \tilde{\chi}_{1,2}^-$) production of 1.6 (0.9) pb at $\sqrt{s} = 13$ TeV, the point remains unchallenged by current LHC results. Recasting with MadAnalysis 5 gives $1 - \mathrm{CL}_s \approx 0.51$ from both the CMS soft leptons [119] and multi-leptons [117] $+ E_T^{\mathrm{miss}}$ searches (CMS-SUS-16-048 and CMS-SUS-16-039), but no constraints can be obtained from simplified model results due to the complexity of the arising signatures. In fact, 86% of the total signal cross section is classified as "missing topologies" in SModelS, i.e. topologies for which no simplified model results are available. The main reason for this is that the $\tilde{\chi}_{3,4}^0$ decay via $\tilde{\chi}_1^\pm$, and thus $\tilde{\chi}_{1,2}^\pm \tilde{\chi}_{3,4}^0$ production gives events with softish jets and/or leptons from 3 off-shell $W$s. It would be interesting to see whether the photons from $\tilde{\chi}_2^0 \rightarrow \tilde{\chi}_1^0 \gamma$ decays would be observable at, e.g., an $e^+ e^-$ collider.

**Point 3 (SPhenoDiracGauginos_12711)** is similar to Point 2 but has a heavier bino-wino mass scale of 560–590 GeV. The $\tilde{\chi}_1^0$–$\tilde{\chi}_2^0$ mass difference is 14 GeV ($\lambda_S = -0.97$) and the LSP is 97% bino and 2% wino. The wino-like states are all compressed within 5 GeV around $m \simeq 590$ GeV. $\Omega h^2 = 0.127$ hence comes dominantly from co-annihilations among the wino-like states, with minor contributions from $\tilde{\chi}_1^0 \tilde{\chi}_2^0 \rightarrow W^+ W^-$ (3%) and $\tilde{\chi}_1^0 \tilde{\chi}_1^\pm \rightarrow WZ$ or $Wh$ (2% each). The collider signatures are, however, quite different from Point 2, given the predominance of photonic decays:

---

[14]This drops to $r \lesssim 0.1$ if displaced $\tilde{\chi}_2^0 \rightarrow \tilde{\chi}_1^0 \gamma$ decays are not explicitly ignored in SModelS.

| mass | decays |
|------|--------|
| 593 GeV | $\tilde{\chi}_2^{\pm} \to \tilde{\chi}_1^{\pm}\gamma$ (77%), $\tilde{\chi}_1^0 W^*$ (23%) |
|  | $\tilde{\chi}_4^0 \to \tilde{\chi}_1^0\gamma$ (61%), $\tilde{\chi}_1^{\pm}W^*$ (27%), $\tilde{\chi}_2^0\gamma$ (7%) |
| 590 GeV | $\tilde{\chi}_3^0 \to \tilde{\chi}_1^0\gamma$ (83%), $\tilde{\chi}_2^0\gamma$ (13%) |
| 588 GeV | $\tilde{\chi}_1^{\pm} \to \tilde{\chi}_2^0 W^*$ (55%), $\tilde{\chi}_1^0 W^*$ (45%) |
| 576 GeV | $\tilde{\chi}_2^0 \to \tilde{\chi}_1^0\gamma$ (92%), $\tilde{\chi}_1^0 Z^*$ (8%); $\Gamma_{\text{tot}} = 3.3 \times 10^{-10}$ GeV (prompt) |
| 562 GeV | $\tilde{\chi}_1^0$, stable |

Moreover, the total relevant EW-ino production cross section is only 41 fb at $\sqrt{s} = 13$ TeV, compared to $\approx 2.6$ pb for Point 2. Therefore, again, no relevant constraints are obtained from the current LHC searches. In particular, SModelS does not give any constraints from EW-ino searches but reports 34 fb as missing topology cross section, 64% of which go on account of $W^*(\to 2$ jets or $l\nu) + \gamma + E_T^{\text{miss}}$ signatures.

**Point 4 (`SPhenoDiracGauginos_2231`)**   has bino and wino masses of the order of 600 GeV similar to Point 3, but features a smaller $\tilde{\chi}_1^0$–$\tilde{\chi}_2^0$ mass difference of 4.5 GeV ($\lambda_S = -0.6$) and a larger spread, of about 46 GeV, in the masses of the wino-like states ($\sqrt{2}\lambda_T = 1.2$). The higgsinos are again heavy. $\Omega h^2 = 0.127$ comes to 46% from $\tilde{\chi}_1^+\tilde{\chi}_1^-$ annihilation; the rest is mostly $\tilde{\chi}_1^{\pm}$ co-annihilation with $\tilde{\chi}_{1,2,3}^0$. The $pp \to \tilde{\chi}_{1,2}^{\pm}\tilde{\chi}_{3,4}^0$ ($\tilde{\chi}_{1,2}^+\tilde{\chi}_{1,2}^-$) production cross section is 24 (12) fb at 13 TeV. Signal events are characterised by multiple soft jets and/or leptons $+E_T^{\text{miss}}$ arising from 3-body decays via off-shell W- or Z- bosons as follows:

| mass | decays |
|------|--------|
| 627 GeV | $\tilde{\chi}_2^{\pm} \to \tilde{\chi}_1^0 W^*$ (62%), $\tilde{\chi}_2^0 W^*$ (9%), $\tilde{\chi}_3^0 W^*$ (20%), $\tilde{\chi}_4^0 W^*$ (7%) |
| 607 GeV | $\tilde{\chi}_4^0 \to \tilde{\chi}_1^{\pm}W^*$ (99.9%) |
| 600 GeV | $\tilde{\chi}_3^0 \to \tilde{\chi}_1^{\pm}W^*$ (99.9%) |
| 581 GeV | $\tilde{\chi}_1^{\pm} \to \tilde{\chi}_1^0 W^*$ (97%), $\tilde{\chi}_2^0 W^*$ (3%) |
| 568 GeV | $\tilde{\chi}_2^0 \to \tilde{\chi}_1^0\gamma$ (98%), $\tilde{\chi}_1^0 Z^*$ (2%); $\Gamma_{\text{tot}} = 3.8 \times 10^{-12}$ GeV (prompt) |
| 564 GeV | $\tilde{\chi}_1^0$, stable |

**Point 5 (`SPhenoDiracGauginos_16420`)**   has the complete EW-ino spectrum below $\approx 700$ GeV. With $m_{DY} < \mu < m_{D2}$ in steps of roughly 100 GeV, the mass ordering is binos < higgsinos < winos. Small $\lambda_S = -0.27$ and large $\sqrt{2}\lambda_T = -0.93$ create small mass splittings within the binos and larger mass splitting within the winos. Concretely, the $\tilde{\chi}_{1,2}^0$ are 99.7% bino-like with masses of 388 GeV and a mass splitting between them of only 200 MeV. The higgsino-like states have masses of about 400–430 GeV and the wino-like ones of about 620–700 GeV. $\Omega h^2 = 0.113$ is dominated by $\tilde{\chi}_1^+\tilde{\chi}_1^-$ annihilation, which makes up 60% of the total annihilation cross section; the largest individual channel is $\tilde{\chi}_1^+\tilde{\chi}_1^- \to Zh$ contributing 14%. Nonetheless $\tilde{\chi}_1^0\tilde{\chi}_1^{\pm}$ (13%) and $\tilde{\chi}_2^0\tilde{\chi}_1^{\pm}$ (12%) co-annihilations are also important. $\tilde{\chi}_1^0\tilde{\chi}_2^0$ co-annihilation contributes about 4%. The decay modes determining the collider signatures are as follows:

| mass | decays |
|---|---|
| 703 GeV | $\tilde{\chi}_3^\pm \to \tilde{\chi}_1^\pm Z$ (78%), $\tilde{\chi}_1^\pm h$ (16%), $\tilde{\chi}_{3,4}^0 W^\pm$ (6%) |
| 670 GeV | $\tilde{\chi}_6^0 \to \tilde{\chi}_4^0 Z$ (45%), $\tilde{\chi}_1^\pm W^\pm$ (36%), $\tilde{\chi}_3^0 h$ (18%) |
| 669 GeV | $\tilde{\chi}_5^0 \to \tilde{\chi}_3^0 Z$ (46%), $\tilde{\chi}_1^\pm W^\pm$ (35%), $\tilde{\chi}_4^0 h$ (18%) |
| 620 GeV | $\tilde{\chi}_2^\pm \to \tilde{\chi}_3^0 W^\pm$ (50%), $\tilde{\chi}_4^0 W^\pm$ (50%) |
| 434 GeV | $\tilde{\chi}_4^0 \to \tilde{\chi}_1^\pm W^*$ (99%) |
| 433 GeV | $\tilde{\chi}_3^0 \to \tilde{\chi}_1^\pm W^*$ (99%) |
| 399 GeV | $\tilde{\chi}_1^\pm \to \tilde{\chi}_2^0 W^*$ (58%), $\tilde{\chi}_1^0 W^*$ (42%) |
| 388 GeV | $\tilde{\chi}_2^0 \to \tilde{\chi}_1^0 \gamma$ (100%); $\Gamma_{\text{tot}} = 4.1 \times 10^{-16}$ GeV ($c\tau \approx 0.5$ m) |
| 388 GeV | $\tilde{\chi}_1^0$, stable |

The $\tilde{\chi}_i^+ \tilde{\chi}_j^-$ and $\tilde{\chi}_i^\pm \tilde{\chi}_k^0$ ($i,j = 1, 2, 3$; $k = 3...6$) production cross sections are 27 fb and 55 fb at the 13 TeV LHC, respectively, but again no relevant constraints can be obtained from re-interpretation of the current SUSY searches.

For the design of dedicated analyses it is relevant to note that $\tilde{\chi}_{2,3}^\pm \tilde{\chi}_{5,6}^0$ production would give signatures like $2W2Z + E_T^{\text{miss}}$ or $3W1Z + E_T^{\text{miss}}$, etc., accompanied by additional jets and/or leptons from intermediate $\tilde{\chi}_{3,4}^0 \to \tilde{\chi}_1^\pm W^*$ decays appearing in the cascade.

We also note that the $\tilde{\chi}_2^0$ is long-lived with a mean decay length of about 0.5 m. However, given the tiny mass difference to the $\tilde{\chi}_1^0$ of 180 MeV, the displaced photon from the $\tilde{\chi}_2^0 \to \tilde{\chi}_1^0 \gamma$ transition will be extremely soft and thus hard, if not impossible, to detect.

**Point 6** (`SPhenoDiracGauginos_11321`)  is a higgsino DM point with $m_{\tilde{\chi}_1^0} \simeq 1.1$ TeV and a rather large mass splitting between the higgsino-like states, $m_{\tilde{\chi}_2^0} - m_{\tilde{\chi}_1^0} \simeq 4$ GeV and $m_{\tilde{\chi}_1^\pm} - m_{\tilde{\chi}_1^0} \simeq 6$ GeV. Here, $\Omega h^2 = 0.112$ results mainly from $\tilde{\chi}_1^0 \tilde{\chi}_2^0$ and $\tilde{\chi}_{1,2}^0 \tilde{\chi}_1^\pm$ co-annihilations. The main decay modes of the heavy EW-ino spectrum are:

| mass | decays |
|---|---|
| 1529 GeV | $\tilde{\chi}_3^\pm \to \tilde{\chi}_1^\pm Z$ (90%), $\tilde{\chi}_1^\pm h$ (8%) |
| 1528 GeV | $\tilde{\chi}_6^0 \to \tilde{\chi}_1^0 Z$ (83%), $\tilde{\chi}_2^0 h$ (6%), $\tilde{\chi}_1^\pm W^\mp$ (7%), $\tilde{\chi}_2^0 Z$ (4%) |
| 1527 GeV | $\tilde{\chi}_5^0 \to \tilde{\chi}_1^0 Z$ (62%), $\tilde{\chi}_2^0 Z$ (22%), $\tilde{\chi}_1^\pm W^\mp$ (8%), $\tilde{\chi}_2^0 h$ (6%) |
| 1526 GeV | $\tilde{\chi}_2^\pm \to \tilde{\chi}_1^\pm Z^\pm$ (60%), $\tilde{\chi}_1^0 W^\pm$ (17%), $\tilde{\chi}_2^0 W^\pm$ (17%), $\tilde{\chi}_1^\pm h$ (6%) |
| 1474 GeV | $\tilde{\chi}_4^0 \to \tilde{\chi}_1^0 Z$ (69%), $\tilde{\chi}_2^0 Z$ (15%), $\tilde{\chi}_1^\pm W^\mp$ (8%), $\tilde{\chi}_2^0 h$ (7%) |
| 1470 GeV | $\tilde{\chi}_3^0 \to \tilde{\chi}_2^0 Z$ (79%), $\tilde{\chi}_1^\pm W^\mp$ (9%), $\tilde{\chi}_1^0 h$ (8%), $\tilde{\chi}_1^0 Z$ (5%) |
| 1081 GeV | $\tilde{\chi}_1^\pm \to \tilde{\chi}_1^0 W^*$ (100%) |
| 1079 GeV | $\tilde{\chi}_2^0 \to \tilde{\chi}_1^0 Z^*$ (89%), $\tilde{\chi}_1^0 \gamma$ (11%); $\Gamma_{\text{tot}} = 9.9 \times 10^{-10}$ GeV (prompt) |
| 1075 GeV | $\tilde{\chi}_1^0$, stable |

The LHC production cross sections are however very low for such heavy EW-inos, below 1 fb at 13–14 TeV. This is clearly a case for the high luminosity (HL) LHC, or a higher-energy machine.

**Point 7** (`SPhenoDiracGauginos_37`)  is another higgsino DM point with $m_{\tilde{\chi}_1^0} \simeq 1.1$ TeV but small, sub-GeV mass splittings between the higgsino-like states, $m_{\tilde{\chi}_2^0} - m_{\tilde{\chi}_1^0} \simeq 120$ MeV and

$m_{\tilde{\chi}_1^\pm} - m_{\tilde{\chi}_1^0} \simeq 400$ MeV. Co-annihilations between $\tilde{\chi}_1^0$, $\tilde{\chi}_2^0$ and $\tilde{\chi}_1^\pm$ result in $\Omega h^2 = 0.124$. The main decay modes are:

| mass | decays |
|------|--------|
| 1957 GeV | $\tilde{\chi}_6^0 \to \tilde{\chi}_1^\pm W^\mp$ (33%), $\tilde{\chi}_{1,2}^0 Z$ (33%), $\tilde{\chi}_{1,2}^0 h$ (31%) |
| 1951 GeV | $\tilde{\chi}_5^0 \to \tilde{\chi}_1^\pm W^\mp$ (33%), $\tilde{\chi}_{1,2}^0 Z$ (32%), $\tilde{\chi}_{1,2}^0 h$ (32%) |
| 1300 GeV | $\tilde{\chi}_3^\pm \to \tilde{\chi}_1^\pm Z$ (55%), $\tilde{\chi}_1^\pm h$ (40%), $\tilde{\chi}_{1,2}^0 W^\pm$ (5%) |
| 1296 GeV | $\tilde{\chi}_{3,4}^0 \to \tilde{\chi}_1^\pm W^\mp$ (44%), $\tilde{\chi}_{1,2}^0 Z$ (31%), $\tilde{\chi}_{1,2}^0 h$ (25%) |
| 1292 GeV | $\tilde{\chi}_2^\pm \to \tilde{\chi}_1^0 W^\pm$ (49%), $\tilde{\chi}_2^0 W^\pm$ (50%) |
| 1159 GeV | $\tilde{\chi}_1^\pm \to \tilde{\chi}_1^0 \pi^\pm$ (69%), $\tilde{\chi}_2^0 \pi^\pm$ (21%); $\Gamma_{\rm tot} = 3.4 \times 10^{-14}$ GeV ($c\tau \approx 6$ mm) |
|  | $\tilde{\chi}_2^0 \to \tilde{\chi}_1^0 \gamma$ (100%); $\Gamma_{\rm tot} = 2.1 \times 10^{-15}$ GeV ($c\tau \approx 92$ mm) |
| 1159 GeV | $\tilde{\chi}_1^0$, stable |

The high degree of compression of the higgsino states causes both the $\tilde{\chi}_2^0$ and the $\tilde{\chi}_1^\pm$ to be long-lived with mean decay lengths of 92 mm and 6 mm, respectively. While the $\tilde{\chi}_2^0$ likely appears as invisible co-LSP, production of $\tilde{\chi}_1^\pm$ (either directly or through decays of heavier EW-inos) can lead to short tracks in the detector. Overall this gives a mix of prompt and displaced signatures as discussed in more detail for Points 9 and 10. Again, cross sections are below 1 fb in $pp$ collisions at 13–14 TeV.

**Point 8 (`SPhenoDiracGauginos_100`)** is the one wino LSP point that our MCMC found (within the parameter space of $m_{DY}, m_{D2}, \mu < 2$ TeV), where the $\tilde{\chi}_1^0$ accounts for all the DM. Three of the wino-like states, $\tilde{\chi}_{1,2}^0$ and $\tilde{\chi}_1^\pm$, are quasi-degenerate at a mass of 1327 GeV, with the forth one, $\tilde{\chi}_2^\pm$, being 5 GeV heavier. The relic density is $\Omega h^2 = 0.11$ as a result of co-annihilations between all four winos. What is special regarding collider signatures is that the $\tilde{\chi}_2^\pm$ decays into $\tilde{\chi}_1^\pm + \gamma$, while the $\tilde{\chi}_1^\pm$ is quasi-stable on collider scales. Chargino-pair and chargino-neutralino production is thus characterised by 1–2 HSCP tracks, in part accompanied by prompt photons. In more detail, the spectrum of decays is:

| mass | decays |
|------|--------|
| 2078 GeV | $\tilde{\chi}_6^0 \to \tilde{\chi}_4^0 Z$ (28%), $\tilde{\chi}_3^0 h$ (21%), $\tilde{\chi}_2^0 h$ (18%), $\tilde{\chi}_1^0 Z$ (14%), $\tilde{\chi}_2^\pm W^\mp$ (10%) |
| 2076 GeV | $\tilde{\chi}_5^0 \to \tilde{\chi}_4^0 h$ (24%), $\tilde{\chi}_3^0 Z$ (24%), $\tilde{\chi}_2^0 Z$ (21%), $\tilde{\chi}_1^0 h$ (12%), $\tilde{\chi}_2^\pm W^\mp$ (11%) |
| 2059 GeV | $\tilde{\chi}_3^\pm \to \tilde{\chi}_3^0 W^\pm$ (41%), $\tilde{\chi}_4^0 W^\pm$ (37%) $\tilde{\chi}_1^\pm Z$ (9%), $\tilde{\chi}_1^\pm h$ (9%) |
| 1356 | $\tilde{\chi}_4^0 \to \tilde{\chi}_1^\pm W^*$ (81%), $\tilde{\chi}_2^\pm W^*$ (19%) |
| 1346 | $\tilde{\chi}_3^0 \to \tilde{\chi}_1^\pm W^*$ (65%), $\tilde{\chi}_2^\pm W^*$ (35%) |
| 1332 GeV | $\tilde{\chi}_2^\pm \to \tilde{\chi}_1^\pm \gamma$ (100%) |
| 1327 GeV | $\tilde{\chi}_1^\pm \to \tilde{\chi}_1^0 e^\pm \nu$ (100%); $\Gamma_{\rm tot} = 2.3 \times 10^{-18}$ GeV ($c\tau \approx 84$ m) |
|  | $\tilde{\chi}_2^0 \to \tilde{\chi}_1^0 \gamma$ (100%); $\Gamma_{\rm tot} = 1.6 \times 10^{-16}$ GeV ($c\tau \approx 1.2$ m) |
| 1327 GeV | $\tilde{\chi}_1^0$, stable |

Like for Points 6 and 7, the LHC cross sections are very low for such a heavy spectrum. Nonetheless **SModelS** gives $r_{\rm max} = 0.28$ from HSCP searches; from the **Pythia**-based recasting we compute $1 - {\rm CL_s} = 0.38$. We hence expect that this point will be testable at Run 3 of the LHC.

**Point 9 (`SPhenoDiracGauginos_625`)** is an example for higgsino-like LSPs at lower mass, around 600 GeV, where the $\tilde{\chi}_1^0$ is underabundant, constituting about 30% of the DM in the standard freeze-out picture. The higgsino-like states are highly compressed, $m_{\tilde{\chi}_1^\pm} - m_{\tilde{\chi}_1^0} \simeq$ 230 MeV and $m_{\tilde{\chi}_2^0} - m_{\tilde{\chi}_1^0} \simeq 435$ MeV, which renders the $\tilde{\chi}_1^\pm$ long-lived with a mean decay length of 55 mm. Direct $\tilde{\chi}_1^\pm$ production has a cross section of about 10 fb at the 13 TeV LHC; more concretely $\sigma(pp \to \chi_1^\pm \chi_{1,2}^0) \simeq 8$ fb and $\sigma(pp \to \chi_1^+ \chi_2^-) \simeq 2$ fb. The $\tilde{\chi}_1^\pm$ can also be produced in decays of heavier EW-inos, in particular of the wino-like $\tilde{\chi}_{3,4}^0$ and $\tilde{\chi}_{2,3}^\pm$, which have masses around 900 GeV. This gives rise to $WZ$, $WH$ and $WW$ events (with or without $E_T^{\rm miss}$) accompanied by short disappearing tracks with a cross section of about 2 fb at 13 TeV. The classic, prompt $WZ$, $WH$, $WW + E_T^{\rm miss}$ signatures also have a cross section of the same order (about 2 fb). While all this is below Run 2 sensitivity, it shows an interesting potential for searches at high luminosity. The detailed spectrum of decays is:

| mass | decays |
|------|--------|
| 1383 GeV | $\tilde{\chi}_6^0 \to \tilde{\chi}_1^\pm W^\mp$ (35%), $\tilde{\chi}_{1,2}^0 Z$ (33%), $\tilde{\chi}_{1,2}^0 h$ (31%) |
| 1381 GeV | $\tilde{\chi}_5^0 \to \tilde{\chi}_1^\pm W^\mp$ (34%), $\tilde{\chi}_{1,2}^0 Z$ (33%), $\tilde{\chi}_{1,2}^0 h$ (32%) |
| 904 GeV | $\tilde{\chi}_3^\pm \to \tilde{\chi}_1^\pm Z$ (49%), $\tilde{\chi}_1^\pm h$ (44%), $\tilde{\chi}_{1,2}^0 W^\pm$ (7%) |
| 901 GeV | $\tilde{\chi}_4^0 \to \tilde{\chi}_{1,2}^0 Z$ (37%), $\tilde{\chi}_{1,2}^0 h$ (31%), $\tilde{\chi}_1^\pm W^\mp$ (33%) |
|  | $\tilde{\chi}_3^0 \to \tilde{\chi}_1^\pm W^\mp$ (34%), $\tilde{\chi}_{1,2}^0 Z$ (33%), $\tilde{\chi}_{1,2}^0 h$ (32%) |
| 898 GeV | $\tilde{\chi}_2^\pm \to \tilde{\chi}_{1,2}^0 W^\pm$ (94%), $\tilde{\chi}_1^\pm h$ (3%), $\tilde{\chi}_1^\pm Z$ (3%) |
| 606 GeV | $\tilde{\chi}_2^0 \to \tilde{\chi}_1^0 \gamma$ (87%), $\tilde{\chi}_1^0 \pi^0$ (11%); $\Gamma_{\rm tot} = 2.5 \times 10^{-13}$ GeV ($c\tau \lesssim 1$ mm) |
|  | $\tilde{\chi}_1^\pm \to \tilde{\chi}_1^0 \pi^\pm$ (96%), $\tilde{\chi}_1^0 l^\pm \nu$ (4%); $\Gamma_{\rm tot} = 3.6 \times 10^{-15}$ GeV ($c\tau \approx 55$ mm) |
| 605 GeV | $\tilde{\chi}_1^0$, stable |

**Point 10 (`SPhenoDiracGauginos_236`)** is another example of a low-mass higgsino LSP point with long-lived charginos. The peculiarity of this point is that the whole EW-ino spectrum lies below 1 TeV: the higgsino-, wino- and bino-like states have masses around 250, 500 and 800 GeV, respectively. The $\tilde{\chi}_1^0$ is highly underabundant in this case, providing only 5% of the DM relic density. Nonetheless the point is interesting from the collider perspective, as it has light masses that escape current limits. Moreover, with a mean decay length of the $\tilde{\chi}_1^\pm$ of about 13 mm, it gives rise to both prompt and DT signatures. Indeed, SModelS reports $r_{\rm max} = 0.39$ for the prompt part of the signal, concretely for $WZ + E_T^{\rm miss}$ from ATLAS-SUSY-2017-03 ($\sigma = 17.51$ fb compared to the 95% CL limit of $\sigma_{95} = 44.97$ fb). The cross section for one or two DTs is estimated as 0.4 pb by SModelS, however the short tracks caused by $\tilde{\chi}_1^\pm$ decays are outside the range of the DT search results considered in section 4.2.2. Last but not least, DTs with additional gauge or Higgs bosons have a cross section of about 50 fb.[15] Recasting with MadAnalysis 5 gives $1 - {\rm CL}_s = 0.73$ (corresponding to $r = 0.6$) from the ATLAS-SUSY-2019-08 [107] analysis. The decay patterns of Point 10 are as follows:

---

[15]See [104, 125] for details on the computation of the prompt and displaced signal fractions in SModelS.

| mass | decays |
|---|---|
| 822 GeV | $\tilde{\chi}_6^0 \to \tilde{\chi}_1^\pm W^\mp$ (35%), $\tilde{\chi}_{1,2}^0 Z$ (34%), $\tilde{\chi}_{1,2}^0 h$ (29%) |
| | $\tilde{\chi}_5^0 \to \tilde{\chi}_1^\pm W^\mp$ (35%), $\tilde{\chi}_{1,2}^0 Z$ (33%), $\tilde{\chi}_{1,2}^0 h$ (30%) |
| 491 GeV | $\tilde{\chi}_3^\pm \to \tilde{\chi}_1^\pm Z$ (50%), $\tilde{\chi}_1^\pm h$ (34%), $\tilde{\chi}_{1,2}^0 W^\pm$ (15%) |
| 486 GeV | $\tilde{\chi}_4^0 \to \tilde{\chi}_{1,2}^0 Z$ (37%), $\tilde{\chi}_1^\pm W^\mp$ (35%), $\tilde{\chi}_{1,2}^0 h$ (28%) |
| 485 GeV | $\tilde{\chi}_3^0 \to \tilde{\chi}_{1,2}^0 Z$ (44%), $\tilde{\chi}_1^\pm W^\mp$ (33%), $\tilde{\chi}_{1,2}^0 h$ (22%) |
| 480 GeV | $\tilde{\chi}_2^\pm \to \tilde{\chi}_{1,2}^0 W^\pm$ (90%), $\tilde{\chi}_1^\pm h$ (5%), $\tilde{\chi}_1^\pm Z$ (5%) |
| 247 GeV | $\tilde{\chi}_1^\pm \to \tilde{\chi}_1^0 \pi^\pm$ (92%), $\tilde{\chi}_1^0 l^\pm \nu$ (8%); $\Gamma_{\rm tot} = 1.5 \times 10^{-14}$ GeV ($c\tau \approx 13$ mm) |
| | $\tilde{\chi}_2^0 \to \tilde{\chi}_1^0 \gamma$ (95%), $\tilde{\chi}_1^0 \pi^0$ (5%); $\Gamma_{\rm tot} = 1.2 \times 10^{-13}$ GeV ($c\tau \approx 2$ mm) |
| 247 GeV | $\tilde{\chi}_1^0$, stable |

**The SLHA files for these 10 points,** which can be used as input for MadGraph, micrOMEGAs or SModelS are available via Zenodo [135]. The main difference between the SLHA files for MadGraph5_aMC@NLO or micrOMEGAs is that the MadGraph5_aMC@NLO ones have complex mixing matrices, while the micrOMEGAs ones have real mixing matrices and thus neutralino masses can have negative sign. The SModelS input files consist of masses, decay tables and cross sections in SLHA format but don't include mixing matrices. The CalcHEP model files for micrOMEGAs are also provided at [135]. The UFO model for MadGraph5_aMC@NLO is available at [84], and the SPheno code at [96].

## 6  Conclusions

Supersymmetric models with Dirac instead of Majorana gaugino masses have distinct phenomenological features. In this paper, we investigated the electroweakino sector of the Minimal Dirac Gaugino Supersymmetric Standard Model. The MDGSSM can be defined as the minimal Dirac gaugino extension of the MSSM: to introduce DG masses, one adjoint chiral superfield is added for each gauge group, but nothing else. The model has an underlying R-symmetry that is explicitly broken in the Higgs sector through a (small) $B_\mu$ term, and new superpotential couplings $\lambda_S$ and $\lambda_T$ of the singlet and triplet fields with the Higgs. The resulting EW-ino sector thus comprises two bino, four wino and three higgsino states, which mix to form six neutralino and three chargino mass eigenstates (as compared to four and two, respectively, in the MSSM) with naturally small mass splittings induced by $\lambda_S$ and $\lambda_T$.

All this has interesting consequences for dark matter and collider phenomenology. We explored the parameter space where the $\tilde{\chi}_1^0$ is a good DM candidate in agreement with relic density and direct detection constraints, updating previous such studies. The collider phenomenology of the emerging DM-motivated scenarios is characterised by the richer EW-ino spectrum as compared to the MSSM, naturally small mass splittings as mentioned above, and the frequent presence of long-lived charginos and/or neutralinos.

We worked out the current LHC constraints on these scenarios by re-interpreting SUSY and LLP searches from ATLAS and CMS, in both a simplified model approach and full recasting using Monte Carlo event simulation. While HSCP and disappearing track searches give quite powerful limits on scenarios with charged LLPs, scenarios with mostly $E_T^{\rm miss}$ signatures remain poorly constrained. Indeed, the prompt SUSY searches only allow the exclusion of (certain)

points with an LSP below 200 GeV, which drops to about 100 GeV when the winos are heavy. This is a stark contrast to the picture for constraints on colourful sparticles, and indicates that this sector of the theory is likely most promising for future work. We provided a set of 10 benchmark points to this end.

We also demonstrated the usefulness of a simplified models approach for EW-inos, in comparing it to a full recasting. While cross section upper limits have the in-built shortcoming of not being able to properly account for complex spectra (where several signals overlap), the results are close enough to give a good estimate of the excluded region. This is particularly true since it is a *much* faster method of obtaining constraints, and the implementation of new results is much more straightforward (and hence more complete and up-to-date). Moreover, the constraining power could easily be improved if more efficiency maps and likelihood information were available and implemented. This holds for both prompt and LLP searches.

We note in this context that, while this study was finalised, ATLAS made `pyhf` likelihood files for the $1l + H(\to b\bar{b}) + E_T^{\mathrm{miss}}$ EW-ino search [107] available on HEPData [136] in addition to digitised acceptance and efficiency maps. We appreciate this very much and are looking forward to using this data in future studies. To go a step further, it would be very interesting if the assumption $m_{\tilde{\chi}_1^\pm} = m_{\tilde{\chi}_2^0}$ could be lifted in the simplified model interpretations.

Furthermore, the implementation in other recasting tools of more analyses with the full $\approx 140$ fb$^{-1}$ integrated luminosity from Run 2 would be of high utility in constraining the EW-ino sector. Here, the recasting of LLP searches is also a high priority, as theories with such particles are very easily constrained, with the limits reaching much higher masses than for searches for promptly decaying particles. A review of available tools for reinterpretation and detailed recommendations for the presentation of results from new physics searches are available in [137].

Last but not least, we note that the automation of the calculation of particle decays when there is little phase space will also be a fruitful avenue for future work.

# Acknowledgements

We thank Geneviève Belanger, Benjamin Fuks, Andre Lessa, Sasha Pukhov, and Wolfgang Waltenberger for helpful discussions related to the tools used in this study. Moreover, we thank Pat Scott for sharing a pre-release version of ColliderBit Solo, and apologise for in the end not using it in this study. MDG would also like to thank his children for stimulating discussions during the lockdown.

**Funding information**  This work was supported in part by the IN2P3 through the projects "Théorie – LHCiTools" (2019) and "Théorie – BSMGA" (2020). This work has also been done within the Labex ILP (reference ANR-10-LABX-63) part of the Idex SUPER, and received financial state aid managed by the Agence Nationale de la Recherche, as part of the programme Investissements d'avenir under the reference ANR-11-IDEX-0004-02, and the Labex "Institut Lagrange de Paris" (ANR-11-IDEX-0004-02, ANR-10-LABX-63) which in particular funded the scholarship of SLW. SLW has also been supported by the Deutsche Forschungsgemeinschaft (DFG, German Research Foundation) under grant 396021762 - TRR 257. MDG acknowledges the support of the Agence Nationale de Recherche grant ANR-15-CE31-0002 "HiggsAutomator." HRG is funded by the Consejo Nacional de Ciencia y Tecnología, CONACyT, scholarship

no. 291169.

## A  Appendices

### A.1  Electroweakinos in the MRSSM

In this appendix we provide a review of the EW-ino sector of the MRSSM in our notation, to contrast with the phenomenology of the MDGSSM.

The MRSSM [19] is characterised by preserving a $U(1)$ R-symmetry even after EWSB. To allow the Higgs fields to obtain vacuum expectation values, they must have vanishing R-charges, and we therefore need to add additional partner fields $R_{u,d}$ so that the higgsinos can obtain a mass (analogous to the $\mu$-term in the MSSM).

| Names | | Spin 0, $R = 0$ | Spin 1/2, $R = -1$ | | $SU(3), SU(2), U(1)_Y$ |
|---|---|---|---|---|---|
| Higgs | $\mathbf{H_u}$ | $(H_u^+, H_u^0)$ | $(\tilde{H}_u^+, \tilde{H}_u^0)$ | | $(\mathbf{1}, \mathbf{2}, 1/2)$ |
| | $\mathbf{H_d}$ | $(H_d^0, H_d^-)$ | $(\tilde{H}_d^0, \tilde{H}_d^-)$ | | $(\mathbf{1}, \mathbf{2}, \text{-}1/2)$ |
| DG-octet | $\mathbf{O}$ | $O$ | $\chi_O$ | | $(\mathbf{8}, \mathbf{1}, 0)$ |
| DG-triplet | $\mathbf{T}$ | $\{T^0, T^\pm\}$ | $\{\chi_T^\pm, \chi_T^0\}$ | | $(\mathbf{1}, \mathbf{3}, 0\ )$ |
| DG-singlet | $\mathbf{S}$ | $S$ | $\chi_S$ | | $(\mathbf{1}, \mathbf{1}, 0\ )$ |
| Names | | Spin 0, $R = 2$ | Spin 1/2, $R = 1$ | Spin 1, $R = 0$ | $SU(3), SU(2), U(1)_Y$ |
| Gluons | $\mathbf{W_{3\alpha}}$ | | $\tilde{g}_\alpha$ | $g$ | $(\mathbf{8}, \mathbf{1}, 0)$ |
| W | $\mathbf{W_{2\alpha}}$ | | $\tilde{W}^\pm, \tilde{W}^0$ | $W^\pm, W^0$ | $(\mathbf{1}, \mathbf{3}, 0)$ |
| B | $\mathbf{W_{1\alpha}}$ | | $\tilde{B}$ | $B$ | $(\mathbf{1}, \mathbf{1}, 0\ )$ |
| R-Higgs | $\mathbf{R_d}$ | $(R_d^+, R_d^0)$ | $(\tilde{R}_d^+, \tilde{R}_d^0)$ | | $(\mathbf{1}, \mathbf{2}, 1/2)$ |
| | $\mathbf{R_u}$ | $(R_u^0, R_u^-)$ | $(\tilde{R}_u^0, \tilde{R}_u^-)$ | | $(\mathbf{1}, \mathbf{2}, \text{-}1/2)$ |

Table 4: Chiral and gauge supermultiplets in the MRSSM, in addition to the quarks and leptons.

The relevant field content is summarised in Table 4. The superpotential of the MRSSM is

$$W^{\text{MRSSM}} = \mu_u\, \mathbf{R_u} \cdot \mathbf{H_u} + \mu_d\, \mathbf{R_d} \cdot \mathbf{H_d} + \lambda_{S_u} \mathbf{S}\, \mathbf{R_u} \cdot \mathbf{H_u} + \lambda_{S_d} \mathbf{S}\, \mathbf{R_d} \cdot \mathbf{H_d}$$
$$+ 2\lambda_{T_u}\, \mathbf{R_u} \cdot \mathbf{T H_u} + 2\lambda_{T_d}\, \mathbf{R_d} \cdot \mathbf{T H_d} \,. \tag{19}$$

Here we define the triplet as

$$\mathbf{T} \equiv \frac{1}{2}\mathbf{T}^a \sigma^a = \frac{1}{2}\begin{pmatrix} \mathbf{T}_0 & \sqrt{2}\mathbf{T}_+ \\ \sqrt{2}\mathbf{T}_- & -\mathbf{T}_0 \end{pmatrix}. \tag{20}$$

Notably the model has an $N = 2$ supersymmetry if

$$\lambda_{S_u} = g_Y/\sqrt{2}, \qquad \lambda_{S_d} = -g_Y/\sqrt{2}, \qquad \lambda_{T_u} = g_2/\sqrt{2}, \qquad \lambda_{T_d} = g_2/\sqrt{2}. \tag{21}$$

The above definitions are common to e.g. [38, 59, 75] and can be translated to the notation of [50] via

$$\lambda_{S_u} \equiv \lambda_u, \qquad \lambda_{S_d} \equiv \lambda_d, \qquad \lambda_{T_u} \equiv \frac{1}{\sqrt{2}}\Lambda_u, \qquad \lambda_{T_d} \equiv \frac{1}{\sqrt{2}}\Lambda_d. \tag{22}$$

The Higgs fields as well as the triplet and singlet scalars have R-charges 0, so their fermionic partners all have R-charge $-1$. The $R_{u,d}$ fields have R-charges 2, so the R-higgsinos have R-charge 1. Together with the "conventional" bino and wino fields, which also have R-charge 1, this gives $2 \times$ four Dirac spinors with opposite R-charges. After EWSB, the EW gauginos and (R-)higgsinos thus form four Dirac neutralinos with mass-matrix

$$
\mathcal{L}_{\mathrm{MRSSM}} \supset -(\tilde{B}, \tilde{W}^0, \tilde{R}_d^0, \tilde{R}_u^0)
\begin{pmatrix}
m_{DY} & 0 & -\frac{1}{2}g_Y v_d & \frac{1}{2}g_Y v_u \\
0 & m_{D2} & \frac{1}{2}g_2 v_d & -\frac{1}{2}g_2 v_u \\
-\frac{1}{2}\lambda_{S_d} v_d & -\frac{1}{2}\lambda_{T_d} v_d & -\mu_d^{\mathrm{eff},+} & 0 \\
\frac{1}{2}\lambda_{S_u} v_u & -\frac{1}{2}\lambda_{T_u} v_u & 0 & \mu_u^{\mathrm{eff},-}
\end{pmatrix}
\begin{pmatrix}
\chi_S^0 \\
\chi_T^0 \\
\tilde{H}_d^0 \\
\tilde{H}_u^0
\end{pmatrix}
\tag{23}
$$

where

$$
\mu_{u,d}^{\mathrm{eff},\pm} \equiv \mu_{u,d} + \frac{1}{\sqrt{2}}\lambda_{S_{u,d}} v_S \pm \frac{1}{\sqrt{2}}\lambda_{T_{u,d}} v_T.
\tag{24}
$$

The above mass matrix looks very similar to that of the MSSM in the case of $N = 2$ supersymmetry!

On the other hand, for the charginos, although there are eight Weyl spinors, these organise into four Dirac spinors, and again into two pairs with opposite R-charges. So we have

$$
\mathcal{L}_{\mathrm{MRSSM}} \supset - (\chi_T^-, \tilde{H}_d^-)
\begin{pmatrix}
g_2 v_T + m_{D2} & \lambda_{T_d} v_d \\
\frac{1}{\sqrt{2}}g_2 v_d & \mu_d^{eff,-}
\end{pmatrix}
\begin{pmatrix}
\tilde{W}^+ \\
\tilde{R}_d^+
\end{pmatrix}
$$

$$
- (\tilde{W}^-, \tilde{R}_u^-)
\begin{pmatrix}
-g_2 v_T + m_{D2} & \frac{1}{\sqrt{2}}g_2 v_u \\
-\lambda_{T_u} v_u & -\mu_u^{eff,+}
\end{pmatrix}
\begin{pmatrix}
\chi_T^+ \\
\tilde{H}_u^+
\end{pmatrix} + h.c.
\tag{25}
$$

The MRSSM therefore does not entail naturally small splittings between EW-ino states. However, if the R-symmetry is broken by a small parameter, then this situation is reversed: small mass splittings would appear between each of the Dirac states.

## A.2   MCMC scan: steps of the implementation

The algorithm starts from a random uniformly drawn point, computes $-\log(L)$ denoted as $-\log(L)_{\mathrm{old}}$, then a new point is drawn from a Gaussian distribution around the previous point, from which $-\log(L)$, denoted as $-\log(L)_{\mathrm{new}}$, is computed. If $pp \times \log(L)_{\mathrm{new}} \leq \log(L)_{\mathrm{old}}$, where $pp$ is a random number between 0 and 1, the old point is replaced by the new one and $-\log(L)_{\mathrm{old}} = -\log(L)_{\mathrm{new}}$. The next points will be drawn from a Gaussian distribution around the point that corresponds to $-\log(L)_{\mathrm{old}}$. The steps of the implementation are the following:

1. Draw a starting point from a random uniform distribution.

2. If point lies within allowed scan range, eq. (6), compute spectrum with SPheno. If the compututation fails, go back to step 1 (or 9).

3. Check if $120 < m_h < 130$ GeV. If not, go back to step 1 (or 9).

4. Call microOMEGAs, check if the point is excluded by LEP mass limits or invisible $Z$ decays, or if the LSP is charged. If yes to any, go back to step 1 (or 9).

5. Compute the relic density and $p_{X1T}$ with microOMEGAs.

6. If relic density below $\Omega h^2_{Planck} + 10\% = 0.132$, save point.

7. Compute $\chi^2_{\Omega h^2}$ for relic density.

8. Compute $-\log(L)_{\text{old}} = \chi^2_{\Omega h^2} - \log(p_{X1T}) + \log(m_{LSP})$.

9. Draw a new point from a Gaussian distribution around the old one.

10. Repeat steps 2 to 7.

11. Compute $-\log(L)_{\text{new}}$.

12. Run the Metropolis–Hastings algorithm:
    $pp$=random.uniform(0,1.)
    **If** $pp \times \log(L)_{\text{new}} \leq \log(L)_{\text{old}}$:
    $\log(L)_{\text{old}}$=$\log(L)_{\text{new}}$

13. iteration++. While iteration$<n_{\text{iterations}}$: repeat steps 9 to 13.

This algorithm was run several times, starting from a different random point each time, to explore the whole parameter space defined by eq. (6).

## A.3  Higgs mass classifier

A common drawback for the efficiency of phenomenological parameter scans, is finding the subset of the parameter space where the Higgs mass $m_h$ is around the experimentally measured value. Our case is not the exception, as $m_h$ depends on all the input variables considered in our study. This is clear for $\mu$, the mass term in the scalar potential, and $\tan\beta$, the ratio between the vevs. For the soft terms, the dependence becomes apparent when one realises that in DG models, the Higgs quartic coupling receives corrections of the form

$$\delta\lambda \sim \mathcal{O}\left(\frac{g_Y m_{DY}}{m_{SR}}\right)^2 + \mathcal{O}\left(\frac{\sqrt{2}\lambda_S m_{DY}}{m_{SR}}\right)^2 + \mathcal{O}\left(\frac{g_2 m_{D2}}{m_{TP}}\right)^2 + \mathcal{O}\left(\frac{\sqrt{2}\lambda_T m_{D2}}{m_{TP}}\right)^2, \qquad (26)$$

where $m_{SR}$ and $m_{TP}$ are the tree-level masses of the singlet and triplet scalars, respectively, and are given large values to avoid a significant suppression on the Higgs mass[16].

To overcome this issue, we have implemented Random Forest Classifiers (RFCs) that predict, from the initial input values, if the parameter point has a $m_h$ inside ($p_{in}$) or outside ($p_{out}$) the desired our $120 < m_h < 130$ GeV range. A sample of 50623 points was chosen so as to have an even distribution of inside/outside range points. The data was then divided as training and test data in a 67:33 split. We trained the classifier using the RFC algorithm in the `scikit-learn` python module with 150 trees in the forest (n_estimators=150).

The obtained mean accuracy score for the trained RFC was 93.75%. However, we are interested in discarding as many points with $m_h$ outside of range as possible while keeping all the $p_{in}$ ones. To do so we have rejected only the points with a 70% estimated probability of being $p_{out}$. In this way, we obtained an improved 98.8% on the accuracy for discarding $p_{out}$ points while still rejecting 86% of them. The cut value of estimated probability for $p_{out}$

---

[16]See for instance, Sec. 2.4 of [69] for a discussion on the effects of electroweak soft terms on the tree-level Higgs mass in DG models.

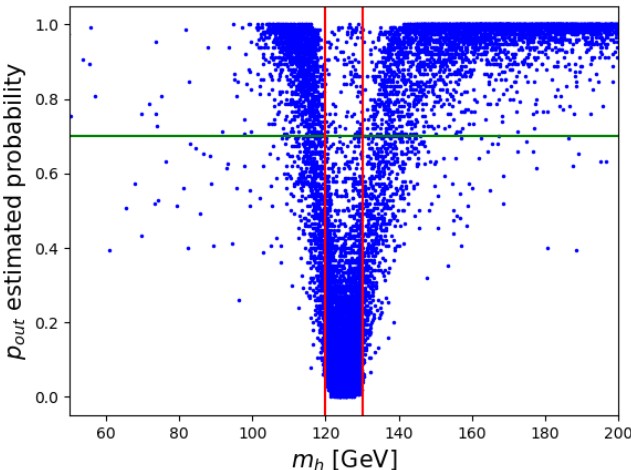

Figure 15: Distribution of the estimated probability for $p_{out}$ as function of $m_h$ obtained from the RFC. Points with an estimated probability above 70% (green line) of being outside the desired $120 < m_h < 130$ range (red lines) are discarded. Values in the $m_h > 200$ GeV and $m_h < 50$ GeV ranges are not depicted for clarity reasons.

was chosen as an approximately optimal balance between accuracy and rejection percentage. Above the 70% value there is no significant improvement in the accuracy, but the rejection percentage depreciates. This behaviour is schematised in Figure 15, where the estimated probability of $p_{out}$ is shown as a function of $m_h$.

Finally, to estimate the overall improvement on the scan efficiency, we multiplied the percentage of real $p_{out}$ (roughly 88%) by the $p_{out}$ rejection percentage (86%) and obtained an overall 75% rejection percentage. Hence, the inclusion of the classifier yields a scan approximately four times faster.

### A.4    Recast of ATLAS-SUSY-2019-08

ATLAS reported a search in final states with $E_T^{\text{miss}}$, 1 lepton ($e$ or $\mu$) and a Higgs boson decaying into $b\bar{b}$, with 139 fb$^{-1}$ in [107]. This is particularly powerful for searching for winos with a lighter LSP (such as a bino or higgsino) and so we implemented a recast of this analysis in MadAnalysis 5 [113–116]. The analysis targets electroweakinos produced in the combination of a chargino and a heavy neutralino, where the neutralino decays by emitting an on-shell Higgs, and the chargino decays by emitting a $W$-boson, i.e. $WH + E_T^{\text{miss}}$. The Higgs is identified by looking for two $b$-jets with an invariant mass in the window $[100, 140]$ GeV, while the $W$-boson is identified through leptonic decays by requiring one signal lepton. Cuts also require $E_T^{\text{miss}} > 240$ GeV, and minimum values of the transverse mass (defined from the lepton transverse momentum and missing transverse momentum). The signal regions are divided into "Low Mass" (LM), "Medium Mass" (MM) and "High Mass" (HM), with four regions for each defined according to the the values of the transverse mass and binned according to the *contransverse mass* of the two $b$-jets

$$m_{\text{CT}} \equiv \sqrt{2 p_{\text{T}}^{b_1} p_{\text{T}}^{b_2} \left(1 + \cos \Delta \phi_{bb}\right)},$$

| $m(\tilde{\chi}_1^{\pm}, \tilde{\chi}_1^0)[\text{GeV}]$ | Region | $m_{\text{CT}} \in [180, 230]$ | | $m_{\text{CT}} \in [230, 280]$ | | $m_{\text{CT}} > 230$ | |
|---|---|---|---|---|---|---|---|
| | | ATLAS | MA | ATLAS | MA | ATLAS | MA |
| $(300, 75)$ | LM | 6 | 7.1 $\pm$2.2 | 11 | 8.5 $\pm$2.5 | 11 | 12.8 $\pm$3.0 |
| $(500, 0)$ | MM | 2.5 | 1.6 $\pm$0.4 | 3.5 | 2.6 $\pm$0.5 | 5.5 | 4.8 $\pm$0.7 |
| $(750, 100)$ | HM | 2 | 2.0 $\pm$0.2 | 2.5 | 2.7 $\pm$0.2 | 6 | 5.4 $\pm$0.3 |

Table 5: Number of events expected in each signal region in [107] (columns labelled "ATLAS") against result from recasting in MadAnalysis 5 (columns labelled "MA") for different parameter points. The quoted error bands are *Monte Carlo* uncertainties, but the cross-section uncertainties can also reach 10% for some regions.

where there are three bins for exclusion limits ($m_{\text{CT}} \in [180, 230]$, $m_{\text{CT}} \in [230, 280]$, $m_{\text{CT}} > 230$) and a "discovery" (disc.) region defined for each $m_T$ region (effectively the sum of the three $m_{\text{CT}}$ bins), making twelve signal regions in all.

This search should be particularly effective when other supersymmetric particles (such as sleptons and additional Higgs fields) are heavy. Given constraints on heavy Higgs sectors and colourful particles, it is rather model independent and difficult to evade in a minimal model. The ATLAS collaboration made available substantial additional data via HEPData at [136], in particular including detailed cutflows and tables for the exclusion curves, which are essential for validating our recast code.

The implementation in MadAnalysis 5 follows the cuts of [107] and implements the lepton isolation and a jet/lepton removal procedure as described in that paper directly in the analysis. Jet reconstruction is performed using fastjet [138] in Delphes 3 [122], where $b$-tagging and lepton/jet reconstruction efficiencies are taken from a standard ATLAS Delphes 3 card used in other recasting analyses [139–142]. The analysis was validated by comparing signals generated for the same MSSM simplified scenario as in [107]: this consists of a degenerate wino-like chargino and heavy neutralino, together with a light bino-like neutralino. The analysis requires two or three signal jets, two of which must be $b$-jets (to target the Higgs decay); the signal is simulated by a hard process of

$$p, p \rightarrow \tilde{\chi}_1^+, \tilde{\chi}_2^0 + n \text{ jets}, \qquad n \leq 2.$$

In the validation, up to 2 hard jets are simulated at leading order in MadGraph5_aMC@NLO, the parton shower is performed in Pythia 8.2, and the jet merging is performed by the MLM algorithm using MadGraph5_aMC@NLO defaults. In addition, to select only leptonic decays of the $W$-boson, and b-quark decays of the Higgs, the branching ratios are modified in the SLHA file (with care that Pythia does not override them with the SM values) and the signal cross-sections weighted accordingly: this improves the efficiency of the simulation by a factor of roughly 8, since the leptonic branching ratio of the $W$ is 0.2157 and the Higgs decays into $b$-quarks 58.3% of the time.

A detailed validation note will be presented elsewhere, including detailed cutflow analysis and a reproduction of the exclusion region with that found in [107]. Here we reproduce the expected (according to the calculated cross-section and experimental integrated luminosity) final number of events passing the cuts for the "exclusive" signal regions, for the three benchmark points where cutflows are available in table 5, where an excellent agreement can be seen. For each point, 30k events were simulated, leading to small but non-negligible Monte-Carlo uncertainties listed in the table.

**Application to the MDGSSM**

To apply this analysis to our model, firstly we treat both the lightest two neutralino states as LSP states; we must also simulate the production of all heavy neutralinos ($\tilde{\chi}^0_i$, $i > 2$) and charginos in pairs. It is no longer reasonable to select only leptonic decays of the $W$, because we can have several processes contributing to the signal. Indeed, in our case, we can have both

$$\tilde{\chi}^+_2 \to \tilde{\chi}^0_{1,2} + W, \tilde{\chi}^0_3 \to \tilde{\chi}^0_{1,2} + H^0$$

and

$$\tilde{\chi}^0_3 \to \tilde{\chi}^-_1 + W, \tilde{\chi}^+_2 \to \tilde{\chi}^+_1 + H^0,$$

for example. Therefore we do not modify the decays of the electroweakinos in the SLHA files, and simulate

$$p, p \to \tilde{\chi}^\pm_{i \geq 1}, \tilde{\chi}^0_{j \geq 3} + n\text{jets}, \qquad n \leq 2$$

as the hard process in MadGraph5_aMC@NLO, before showering with Pythia 8.2 and passing to the analysis as before.

We have not produced an exclusion contour plot for this analysis comparable to the MSSM case in [107], because a heavy wino with a light bino always leads to an excess of dark matter unless the bino is near a resonance. We should generally expect the reach of the exclusion to be better than for the MSSM, due to the increase in cross section from pseudo-Dirac states; since we can only compare our results directly for points on the Higgs-funnel, for $m_{\tilde{\chi}_1} \approx m_h/2$, we find a limit on the heavy wino mass of about 800 GeV in our model, compared to 740 GeV in the MSSM.

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
