# Peer review of "Constraining Electroweakinos in the Minimal Dirac Gaugino Model"

_SciPost Physics_

## Round 1 · Referee Report · Anonymous · 2020-8-5

Report
The article describes a recast of exisiting experimental LHC studies to derive bounds on supersymmetric models with Dirac gauginos. Specifically, the authors consider the Minimal Dirac Gaugino SUSY Model, which is maybe not the best motivated model from a theoretical point of view, but it may serve as a minimal implementation of the essential features that are shared by a range of supersymmetric Dirac gaugino models. The present work builds on earlier work by the same authors and others. Its main useful conclusions beyond the existing literature are:
It was found that a fast and simplified check of the excluded parameter space from LHC searches using SModelS is mostly adequate, and costly MC simulations are for the most part not needed. This is important since the MDGSSM is only one of many SUSY-like BSM models, and it would be impractical to run MC simulations for a recast of each of these.
The inclusion of LLP constraints has a significant impact on the parameter space that can be probed by LHC experiments. It is not an entirely new insight that LLP bounds are important for probing DM models, but the treatment of these signatures in a mostly automated and efficient computer framework is very useful. I suppose this work can be considered as a demonstration that SModelS can deliver such a framework, which can be adapted by researchers for testing their favorite model.
The authors provide a set of benchmark points which could be helpful for the design of future experimental studies at the LHC experiments. However, the choice of these benchmark points is not fully explained (see below), as some of them appear to offer little incentive for the development for new analysis methods, while on the other hand it is not explained if the 10 proposed points are characteristic of all regions of the full model parameter space.
These items generally meet the SciPost acceptance criteria, since it opens a pathway for future research and its presentation is overall clear and detailed.
However, before the article can be recommended for publication, I ask the authors to consider the following questions:
It is not clear how in what sense Fig.1 displays a comparison to Refs. 92-94. As far as I can see it only shows curves computed in the present work. Also, the meaning of the red curve is not explained.
I'm not sure if I understand the discussion at the end of section 3.2.2. The loop functions x_1 through x_4, when computed explicitly, should reflect the Ward identity relations in (16). Thus when plugging in these loop functions in (17) or in (18), one should manifestly obtain the same result.
Fig.3(b): Contrary to what is written in the text, I cannot see any wino-like points.
What is the meaning of the r_{max} > 0.5 points in Figs. 10 and 11?
The authors use SModelS for the analysis of HSCP limits from Refs.[118,119]. However, they use a different recast code and MC simulations for deriving HSCP limits from [121], requiring significantly more computing resources. Why was it not possible to implement the results from [121] in SModelS?
I do not quite understand the rationale for choosing the benchmark points in section 5. Benchmark points can be very valuable for helping to design experimental studies to extend their coverage and capture previous overlooked signatures. However, points 1 and 6 appear to predict fairly standard signatures that are already pretty well covered by existing LHC searches (and their future extensions to larger data samples). Point 6 will be difficult to probe at LHC simply due to lack of statistics, but otherwise does not require the development of new analysis techniques. Points 7 and 9 both call for a mix of prompt multi-boson and LLP signatures. It is not clear what is the crucial distinction between these two points.
The manuscript fails to cite some well-known previous work on the LHC phenomenology of Dirac gauginos, such as the arXiv numbers
0808.2410, 0902.3795, 1005.0818, 1307.7197, 1803.00624

---

## Round 1 · Referee Report · Anonymous · 2020-8-18

Report
The authors studied the electronweakino sector in the MDGSSM with respect to dark matter and collider constraints. In particular, they considered the regions of parameter space where the lightest neutralino accounts for at least part of the DM relic abundance while being consistent with the existing bounds from DM direct detection, LEP and low-energy data and LHC Higgs measurements. They then constrain the model points in those parameter regions using the existing relevant LHC searches with the aid of two packages SModelS and MadAnalysis 5. As many of the model points contain long-lived particles, they performed their analysis, considering the widths of those decays carefully.
The manuscript is well written and provides useful information including the benchmark points that they proposed, in terms of investigating the MDGSSM at colliders. However, I would bring a few comments/suggestions of mine to authors’ attention and ask them to address those comments before being published in SciPost Physics.
Requested changes
1) Below eq. (8), the authors allowed for a 10% theoretical uncertainty, but I couldn’t find the reason for that. It would be great to briefly explain their choice and potential impacts of different choices on their final results.
2) Right before section 3.2, the authors stated that 52,550 scan points pass the constraints described in section 3.1. I was wondering how many points were tested. Since there are six parameters to vary according to eq. (6), if just 10 different values were tried for each parameter, I would expect 10^6 scan points to be tested. This information would help readers develop the intuition what fraction of the parameter space would be excluded or not.
3) At the beginning of section 3.2.1, the authors stated that when \Delta m <1.5 GeV, it is not accurate to describe the W* decays in terms of quarks. Just for more interested readers, I would recommend the authors to refer to relevant literature or provide a brief discussion for that.
4) It seems there’s a white band in-between c\tau = ~10^3 mm and ~10^4 mm in Figure 13. It would be good to add comments for that white band.
5) Finally, there are several typos here and there, for example, nucleii -> nuclei in pape 3, \tau meson -> \tau lepton in page 8, single pions -> a single pion in page 11, so I would the authors to go over the manuscript carefully to correct those typos.

---

## Round 2 · Referee Report · Anonymous (Referee 2) · 2020-9-7

Report

The authors have addressed all my comments, so I recommend this manuscript for publication.

---

## Round 2 · Referee Report · Anonymous (Referee 1) · 2020-9-8

Report

The revised version of the manuscript and the author's response has clarified several of my previous questions. However, there are a few outstanding issues or misunderstandings:

The authors have not addressed my previous question about the meaning of the Ward identity. My point is the following: In contrast to what is written in the text, both when using eq.(17) or (18) one should get the same result. The only exception is when there is an error in the calculation of any of the $x_{1,2,3,4}$. However, when $x_{1,2,3,4}$ are computed correctly, they should explicitly satisfy eq.(16), and thus the result would not change whether eq.(16) is applied or not.

Concerning Fig.3(b), maybe the location of the wino-like point could be indicated by a circle or arrow?

I do not quite agree with the rationale for choosing benchmark points. In their response, the authors state that they are useful for "investigating how and to what extent the EW-ino sector of the MDGSSM may be tested at colliders." However, a set of 10 benchmark points cannot reasonably do that, but this rather requires a broad parameter scan (which the author do in fact carry out in the previous sections of this paper). In the end, the choice of benchmark points will always include some subjectivity, of course, but I think there is a benefit to limiting them to a minimal number of characteristic scenarios.

---

## Round 2 · Author Response

We thank the two referees for their careful and positive assessment of our paper. Our replies and modifications to the manuscript are as follows:

Referee 1:

  1. It is not clear how in what sense Fig.1 displays a comparison to Refs. 92-94. As far as I can see it only shows curves computed in the present work. Also, the meaning of the red curve is not explained.

REPLY: We have rephrased the relevant passage in the text and added a labels for the red curves in Figs. 1 and 2.

  1. I'm not sure if I understand the discussion at the end of section 3.2.2. The loop functions x_1 through x_4, when computed explicitly, should reflect the Ward identity relations in (16). Thus when plugging in these loop functions in (17) or in (18), one should manifestly obtain the same result.

REPLY: Yes; equation (18) is the result of inserting relations (16) into equation (17).

  1. Fig.3(b): Contrary to what is written in the text, I cannot see any wino-like points.

REPLY: There is one yellow point with about 95% wino content. It is true it is hard to spot, but it becomes clear when looking also at Fig. 4(b).

  1. What is the meaning of the r_{max} > 0.5 points in Figs. 10 and 11?

REPLY: This is primarily to illustrate how the reach might improve with, e.g., more statistics. But, it also roughly indicates the effect of a possible underestimation of the visible signal in the simplified model approach. We have added a paragraph below Figure 11 to explain this.

  1. The authors use SModelS for the analysis of HSCP limits from Refs.[118,119]. However, they use a different recast code and MC simulations for deriving HSCP limits from [121], requiring significantly more computing resources. Why was it not possible to implement the results from [121] in SModelS?

REPLY: Ref [121] (now [126]) provides cross section upper limits for chargino pair production, but these assume that the charginos are quasi-stable. To use these limits in SModelS, we’d still need to emulate the lifetime and boost dependence. Indeed, to implement the results from [121] on the same footing as those of [118,119] (now [123,124]) in SModelS, one should develop the necessary HSCP efficiency maps from MC simulations similar to what was done in 1808.05229. This is a heavy work involving even more computing resources than the simulations we’ve done. Besides, we found the comparison with MC-based recasting interesting and relevant. An update of SModelS with new LLP/HSCP results is left to future work of the SModelS collaboration.

  1. I do not quite understand the rationale for choosing the benchmark points in section 5. Benchmark points can be very valuable for helping to design experimental studies to extend their coverage and capture previous overlooked signatures. However, points 1 and 6 appear to predict fairly standard signatures that are already pretty well covered by existing LHC searches (and their future extensions to larger data samples). Point 6 will be difficult to probe at LHC simply due to lack of statistics, but otherwise does not require the development of new analysis techniques. Points 7 and 9 both call for a mix of prompt multi-boson and LLP signatures. It is not clear what is the crucial distinction between these two points.

REPLY: The 10 benchmark points are characteristic for the kind of spectra that appear, illustrating differences but also common features with the MSSM. Their interest lies not only in “helping to design experimental studies to extend their coverage and capture previous overlooked signatures” at the LHC, but more generally in investigating how and to what extent the EW-ino sector of the MDGSSM may be tested at colliders (LHC, ILC, FCC, or whatever). This also includes studies of the inverse problem and of ways to distinguish extended models from the MSSM in case of a discovery, etc.. We think that our set of benchmark points is pertinent and useful in these respects. By the way, the 2nd referee agrees with this view.

  1. The manuscript fails to cite some well-known previous work on the LHC phenomenology of Dirac gauginos, such as the arXiv numbers 0808.2410, 0902.3795, 1005.0818, 1307.7197, 1803.00624

REPLY: We thank the referee for pointing out these missing references. We are now citing them in the introduction.

Referee 2:

  1. Below eq. (8), the authors allowed for a 10% theoretical uncertainty, but I couldn’t find the reason for that. It would be great to briefly explain their choice and potential impacts of different choices on their final results.

REPLY: This is a rough estimate to account e.g. for the fact that the relic density calculation is done at the tree level only. 10% uncertainty from higher order effects seem reasonable and avoid an extreme fine-tuning of parameters. We have added a clarifying remark (in parentheses) below eq. (8).

  1. Right before section 3.2, the authors stated that 52,550 scan points pass the constraints described in section 3.1. I was wondering how many points were tested. Since there are six parameters to vary according to eq. (6), if just 10 different values were tried for each parameter, I would expect 10^6 scan points to be tested. This information would help readers develop the intuition what fraction of the parameter space would be excluded or not.

REPLY: Indeed some 10^6 points were tested. We have added this information in the text. We like to point out, however, that the fraction of excluded parameter space depends a lot on the sampling method.

  1. At the beginning of section 3.2.1, the authors stated that when Delta m <1.5 GeV, it is not accurate to describe the W* decays in terms of quarks. Just for more interested readers, I would recommend the authors to refer to relevant literature or provide a brief discussion for that.

REPLY: We now refer the reader to Appendix A of hep-ph/9607421, where it is pointed out that “Standard expressions for chi^pm -> chi^0 + f bar f are not applicable for such small mass differences, since they assume the final state fermions to be massless. Moreover, hadronic decays can only be described by perturbative QCD if the mass difference exceeds one or two GeV.”

  1. It seems there’s a white band in-between ctau = ~10^3 mm and ~10^4 mm in Figure 13. It would be good to add comments for that white band.

REPLY: This white band results from an abrupt change in the chargino lifetime as decays into pions become kinematically forbidden. We have added a clarifying remark in the discussion of Figure 13.

  1. Finally, there are several typos here and there, for example, nucleii -> nuclei in pape 3, tau meson -> tau lepton in page 8, single pions -> a single pion in page 11, so I would the authors to go over the manuscript carefully to correct those typos.

REPLY: We have fixed these and a few other typos. “single pions” on page 11 is, however, correct to our mind (because neutralinos is also plural).

We hope that these replies and modifications to the manuscript are satisfactory, and that our paper is now ready for publication in SciPost Physics.

---

## Round 2 · List of Changes

See replies to referee comments.

---

## Editorial Decision

resubmitted